


# The quest for reference stations at the National Observatory of Athens, Greece

Olga-Joan Ktenidou[1], Antonia Papageorgiou[12], Erion-Vasilis Pikoulis[12], Spyros Liakopoulos[1], Fevronia Gkika[1], Ziya Cekinmez[13], Panagiotis Savvaidis[14], Kalliopi Fragouli[15], Christos Evangelidis[1]

[1]Institute of Geodynamics, National Observatory of Athens, Lofos Nymfon-Thiseion, Athens, 11850, Greece
[2]University of Patras, Patras, 26504, Greece
[3]National Technical University of Athens, Athens, 15780, Greece
[4]Aristotle University of Thessaloniki, Thessaloniki, 54124, Greece
[5]University of the Aegean, Mytilene, 22510, Greece

Correspondence to: Olga-Joan Ktenidou (olga.ktenidou@noa.gr)

**Abstract.** The assumption of reference station conditions is investigated for the first time across 60 rock stations belonging to the broadband and accelerometric networks of the National Observatory of Athens. We select the stations based on the established belief that they lie on rock, and provided that their data have been publicly available for long enough to yield a substantial number of recordings. No site effects studies have been conducted before for the ensemble of the stations under study. Furthermore, no ad hoc field campaigns have been performed to characterise them, save in few cases. The first step is to compile all existing information for these stations from all publicly available sources and past studies, including geology, topography, station installation, $Vs_{30}$ estimates and any other known metadata. The second step is to compile ad-hoc information from maps combined with the operator's first-hand experience of the sites, to better describe the geological unit and age, along with other characteristics such as station installation and morphology. The third and largest step is to compile the first Greek ground-motion dataset on rock and to perform a detailed analysis of the recordings to estimate site-specific transfer functions and hence assess local site response characteristics for each station. A strong-motion dataset of 6840 recordings is developed and curated for this purpose, visually inspected and processed in the time and frequency domains. Single-station amplification functions (horizontal-to-vertical spectral ratios, HVSR) are estimated from the seismic data, and the site resonance characteristics are assessed, not only in the conventional way of combining horizontal components, but also assessing the transfer function's directional sensitivity. Considering that 'true' reference site behaviour implies low, flat amplification with no directional dependence, these transfer function characteristics are combined with the compiled station metadata -existing and new- to evaluate the stations' overall capacity as reference sites. The stations are grouped in terms of behaviour and the preferred ones are recommended, hoping to facilitate the better use of seismic data in future hazard applications.


## 1 Introduction

The importance of understanding site conditions at strong-motion recording stations has been known for decades. Important
global databases such as NGA-West2 (Ancheta et al., 2014) made a point of procuring rich and homogeneous station
metadata in terms of Vs, depth to bedrock, etc. Ground motion models have moved towards more detailed descriptors of
station conditions, and a global effort is being made in characterising stations. In recent years, particular importance has been
attached to assessing ground motion on rock sites in particular, while in the past it was considered as rather homogeneous
(some notable exceptions including the seminal works of Silva & Darragh 1995 and Steidl 1996). We now recognise that
material properties and geometry –the main ingredients of site response- can cause ground motions to differ strongly
between rock stations, and that they are not as 'uninteresting' as we once thought in terms of site response (i.e., the implicit
assumption of negligible amplification dos not hold). This has important potential impact on reference ground motions and
the definition of reference stations, which once were simply defined as those coming from 'rock' sites. It has impacted
seismic hazard and risk assessment for significant structures and critical infrastructures, which now often accounts in detail
for such rock property variations. However, rock sites can be notoriously challenging to characterise, and many networks
have not characterised their rock stations, as priority had been initially –and reasonably- given to stations lying on soils.

Some studies in the past decade or so attempted to focus on rock sites. Van Houtte et al. (2012) tested stations in
Christchurch that were typically used as reference stations without previous checks, by computing site transfer functions.
Ktenidou & Abrahamson (2016) found broadband amplifications even in CENA rock sites that had been considered as
extremely hard ($Vs_{30}$ of 2000 m/s). More recently, much more systematic and large-scale efforts have been made on
European level by Lanzano et al. (2020), who made a large-scale detailed effort for defining reference sites in Central Italy
using various proxies as well as transfer functions from seismic data and noise, as did Pilz et al. (2020) who also included
artificial intelligence tools in their reference site identification. Di Giulio et al. (2021) attempted to assess in a systematic
way the seismic station characterisation efforts across Europe in terms of data quality, methodological reliability etc.,
emphasising the importance of consistency.

In Greece, whose seismic data are of great importance to European and even global ground-motion datasets, relatively little
progress has been made so far in characterising stations. Many logistical reasons lie behind this, including the fact that a
significant number of seismic networks are run by different operators exist (Evangeldis et al., 2021), there is a large number
of stations off the mainland or in areas that are difficult to approach due to terrain, etc. Some efforts have been made to
60 compile what station metadata exist, since the early days of HEAD, the first strong-motion database (Theodoulidis et al.,
2004). Margaris et al. (2014) provided a brief history of the characterisation of Greek strong-motion stations with boreholes,
geophysical campaigns and microtremors, while Stewart et al. (2014) compiled values of $Vs_{30}$ and other site descriptors for
some strong-motion stations, mostly based on information within a 1-km radius from the stations per se. Margaris et al.
(2021) include the most up-to-date version of available strong-motion station metadata, mostly through proxies. We note that
the ensemble of stations considered in all the above studies includes a large number of stations that lie on soft ground, and a



large fraction of them are not yet publicly available through European waveform services (ESM). Only one systematic effort was made so far, for one of the Greek networks (HI, doi:10.7914/SN/HI) by Grendas et al. (2018), in which the actual strong-motion recordings were analysed to compute empirical transfer functions to understand site amplification; however, the majority of those stations are again not publicly available in terms of waveform data.

The goal of this work is to focus on the networks of the National Observatory of Athens (doi:10.7914/SN/HL), including not only the strong-motion one (https://accelnet.gein.noa.gr) but also the broadband seismic one (https://bbnet.gein.noa.gr/HL/), and further focus on the stations openly available in real-time continuous mode through the EIDA@NOA node (Evangelidis et al., 2021). For a fraction of the strong-motion stations, site conditions are known in great detail thanks to geophysical in situ investigations conducted in the recent national project HELPOS (Hellenic Plate Observing System); however, most of

these stations are either not open or lie on soils. To date, most of the openly available strong-motion stations are still characterized via proxies, while none of them have been analysed to determine empirical amplification functions (spectral ratios). Moreover, there has never been a systematic, consistent effort to include broadband stations as well, despite the increasing importance that is recently being attached to broadband data in ground-motion databases. In the HL networks, only a few small-scale efforts were made in the recent past to understand the behaviour of selected strong-motion and

broadband stations using the recordings themselves (Ktenidou & Kalogeras, 2019; Ktenidou et al., 2021a, 2021b). These were made using only limited datasets, mostly as proof of concept to the work at hand. This paper marks the beginning of a more systematic study of the NOA network conditions, starting with rock sites.

## 2 Strong-motion data and analysis

### 2.1 Station and data selection

All stand-alone broadband stations (HH channels) and collocated broadband and strong-motion stations (HH and HN channels) are generally thought to lie on rock conditions. Hence all such stations are included in this study, as long as they had enough recordings at the end of 2021, which could be publicly accessible via the EIDA@NOA node at that time (Evangelidis et al., 2021). In addition, stand-alone strong-motion stations (HN) open to the public via EIDA were considered, and those thought to lie on rock were selected. The layout of the stations selected is shown in Fig. 1b, and some

basic information about them is compiled in Table 1 (where 'HNc' indicates strong-motions stations installed at the same site as a broadband station).

A threshold minimum magnitude of ML4 was considered for each station, dropping down to M3.5 only in one case, for a station installed in 2021. The maximum distances considered varied according to noise level and station population of recordings, but scaled from out to 150 km for smaller events and out to 300 km or more for large events. The overall M-R

distribution is shown in Fig. 2 for the ensemble dataset, and can be found on a station-specific basis in the Supplement (Figs S1 and S2). Because the purpose of this dataset is the study of site effects (not, for instance, the development of ground motion models) and the M-R distributions are used as an indication only, we use local magnitude scale and epicentral


distance metrics and do not go into the details of moment magnitude and rupture distance for the large events in the dataset.
A total of 6840 recordings are analysed in this study. The number of records per station is shown in Fig. 2b. The minimum

number of usable recordings for the single least populated station is 8, the mean number of recordings is 90, while some
stations have more than 300.

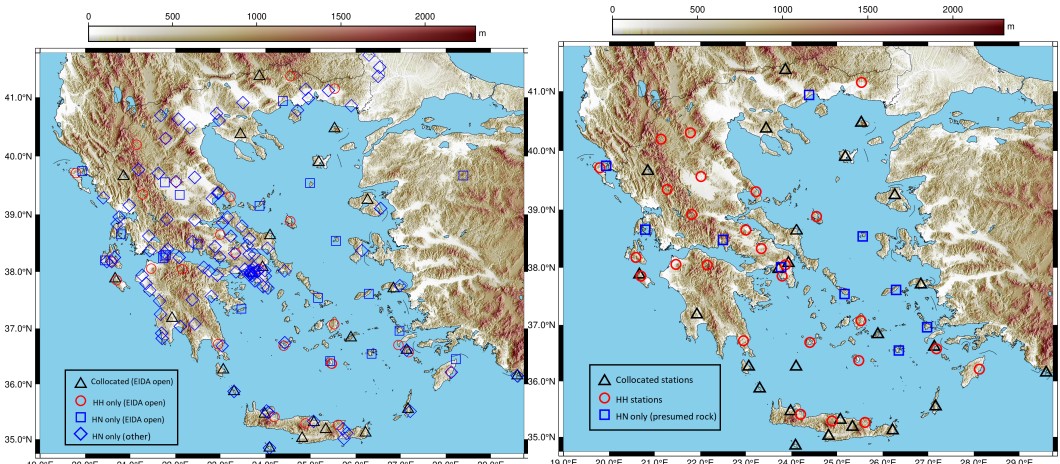

**Figure 1: a.** Map of all HL stations in the end of 2021. **b.** Map of selected stations (believed to lie on rock, with publicly available
data via EIDA@NOA and adequate number of events).

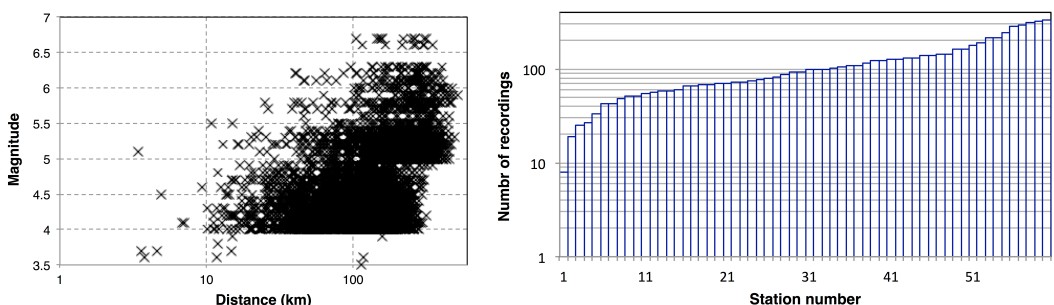

**Figure 2:** a. Indicative distribution of magnitude (local) and distance (epicentral) for all data analysed in this study. Station-
specific plots can be found in the Supplement (Figs S1 and S2). b. Number of recordings used per station.




### 2.2 Creation of a new strong-motion dataset

The data we select come from the period 2012-2021, depending on when each station began to operate in real-time, its period of operation and data availability. We use the catalogue of NOA (https://eida.gein.noa.gr/fdsnws/availability/1) and search for recordings following the criteria mentioned above. We retrieve raw waveforms and station xml from EIDA@NOA and apply instrument correction to retrieve physical units. We check raw HH data for clipping and discard all

115 such instances. We use an in-house software that follows closely upon the rationale described in Kishida et al. (2016), the procedure that underpins the NGA-East processing (Goulet et al., 2014). We first perform visual inspection in the time domain (Fig. 3a), where the windowing has been automatically done based on the origin time (P and S arrivals, selection of equal duration pre-event noise and signal windows). The signal window includes all wave packages of engineering interest, i.e. all S waves and the most energetic surface waves. All automatic picks are assessed and corrected as necessary. We then

perform visual inspection in the frequency domain (Fig. 3a), assessing smoothed and unsmoothed S-wave and noise Fourier amplitude spectra (FAS) of acceleration. Aside from the signal-to-noise ratio (SNR=3 threshold), we also consider the fit to the omega-square source model (Brune 1970; 1971). Figure 3 shows an example of window selection in the time domain, and of the lowest and highest usable frequencies (LUF, HUF) in log and linear scale respectively. All FAS will be used within their usable frequency to compute the empirical transfer functions in what follows.

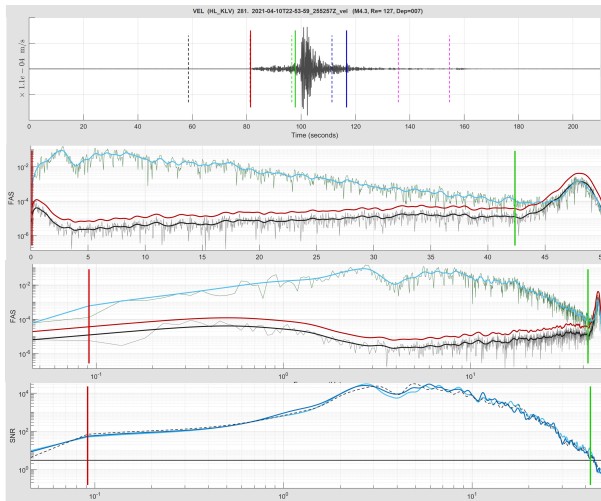

**Figure 3:** Top to bottom: Example manual processing: windowing of a velocity trace in the time domain, selecting the HUF and LUF in the frequency domain (in linear and log scale respectively) for the two horizontal acceleration FAS, and inspection of the SNR.



### 2.3 Transfer functions

The start of the S-wave windows is taken early enough so as for the first S waves not to be affected by the tapering. The acceleration FAS are computed and smoothed with a Konno & Ohmachi (1998) b=40 mild smoothing. We compute the horizontal-to-vertical spectral ratio (HVSR; Lermo & Chavez-Garcia, 1993) for each component of each recording at each

135 station. The mean HVSR per site is computed as the log average across all events, as is customary, and as Ktenidou et al. (2011) showed that spectral ratio ordinates are lognormally distributed. At each frequency, the mean is computed out of the available recordings within the legitimate bandwidth. Within the range of 1-10 Hz, typically all recordings are usable, which as noise increases towards lower and higher frequencies, fewer recordings are strong enough to contribute. The two components of each FAS are combined as the square root of the sum of squares (SRSS) so as to yield an orientation-

140 independent estimate. Fig. 4 (top) shows two examples of this mean HVSR ±1 SD. We note that the curves are only drawn where the number of usable events is at least 5, in order to ensure a more robust estimate of the statistics (most ground motion applications will accept a minimum of 3). Figures S4 and S5 in the Supplement show results for all HH and HN stations respectively.
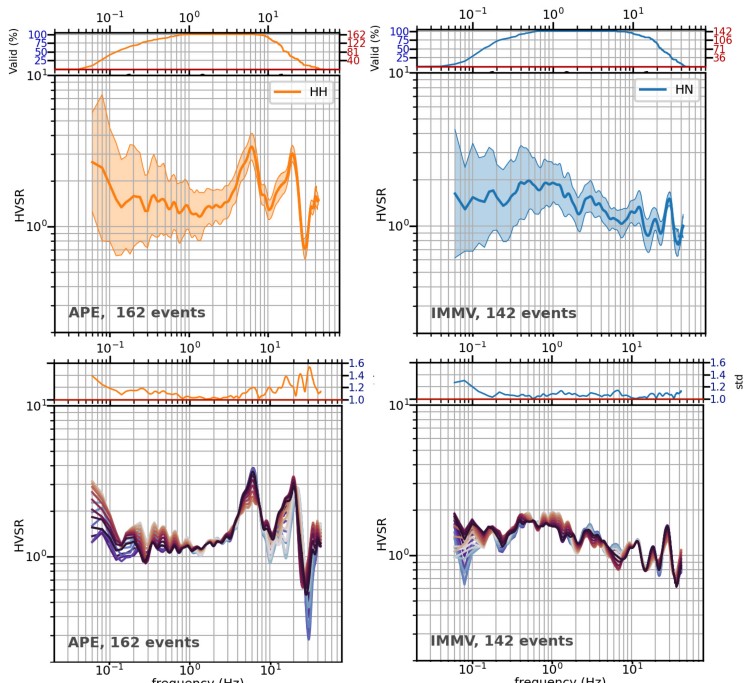

**Figure 4:** Example HVSR results for stations APE.HH (left) and IMMV.HN (right). Top row: mean, direction-invariant (SRSS) HVSR ±1 standard deviation; inset on top indicates the number and percentage of usable recordings per frequency. Bottom row: HVSR per component, as those are rotated by 10-degree intervals from North to East; inset on top indicates the standard deviation (hence, directional sensitivity or variability) per frequency.

A reference site is expected to exhibit a HVSR that is relatively flat and close to unity. Departure from reference site conditions has been judged in different ways across different studies. A few example thresholds include the typical value of HVSR>2, but also HVSR>2√2 (Lanzano et al., 2020 from Puglia et al., 2011), and the more generous one of HVSR>3 (Pilz et al., 2020). Of course, HVSR is an approximation, and generally an underestimation with respect to the 'true' site transfer function, for instance as that may be computed using the standard spectral ratio (SSR) of Borcherdt (1970), i.e., using an actual rock recording as reference rather than the vertical. The assumed premise of HVSR should not be that the vertical component actually remains unaltered by stratigraphy (or, indeed, by other geomorphological features), but rather than it is expected to exhibit amplification at frequencies higher than the ones the horizontal ground motion amplifies around, thus permitting a rather clear identification of at last the first resonant peak, albeit at a generally lower level than the actual. It is


for this reason that we take the stricter view of a threshold of HVSR>2 when attempting to identify potential reference sites
among our group of 60 stations.

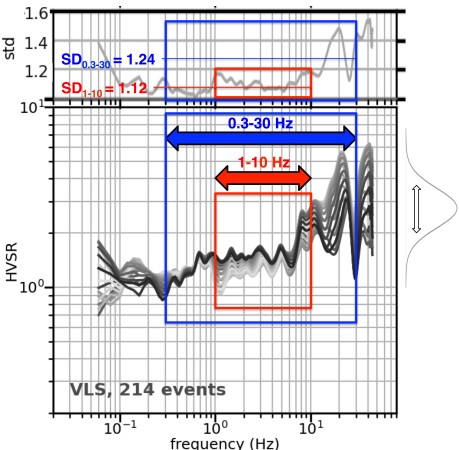

**Figure 5:** Illustration of the two frequency bands over which the standard deviation from the rotations is averaged, to derive
an index of directional variability: 0.3-30 Hz (blue) and 1-10 Hz (red). For station VLS, the value is low for the narrow band
(1.12) but high for the wider one (1.24) due to high-frequency variability.

We also expect a reference site to not exhibit strong directional dependence, i.e., reference ground motions not to be
sensitive to the sensor installation orientation. However, checking only the difference between the two horizontal
components as installed is not rigorous enough. The sensors are installed in the N and E directions, which are arbitrary with
respect to each site's potential geomorphological features. This is why we follow the technique of Ktenidou et al. (2016) to
assess the variability of site response to azimuth. We rotate each time series by successive increments of 10°, from 0°-90°,
and recompute the FAS and HVSR each time, so as to discover whether there are any other directions that may bring out
directional differences. Such differences we view as an indication of departure from 1D behaviour due to local
geomorphology (basin edges, topography, lateral discontinuities, etc.). All of these factors can cause amplification of
different levels in the two horizontal components, e.g. the radial and transverse with respect to the feature's axis. Fig. 4
(bottom) shows two examples of the mean HVSR per component, as the as-installed motions are rotated by 10-degree
increments from North to East. The inset on top indicates the standard deviation of the man HVSR values across all rotations
per frequency. We consider this as an index of the directional variability of each station's site response. Though the typical
parameters extracted from such calculations are most of all the resonant frequency $f_0$, and –to some extent of credibility,
mostly as an indication- the corresponding amplitude $A_0$ and perhaps the same metrics for the first higher mode, if

applicable, we also take note of the directional variability of the transfer function amplitude. To this end, we compute the mean of this variability function with frequency across two indicative ranges of interest, namely a wide one spanning 2 orders of magnitude (0.3-30 Hz) and a narrower one of 1 order of magnitude, which may also be more interesting for typical

structural response (1-10 Hz). We also note the value of this function around the resonant frequency of the site. We propose that these three values ($SD_{0.3-30}$, $SD_{1-10}$, $SD_{fo}$) can be used as approximate indicators of the azimuthal stability of site response. Figure 5 illustrates these values for station VLS, where such scatter begins above 10 Hz and thus affects mostly $SD_{0.3-30}$ (1.24) and $SD_{fo}$ (1.48 at 20 Hz). The value of $SD_{1-10}$ (1.12) is relatively low for this dataset. In the examples of Fig. 4, station APE exhibits non-negligible directional variability around its $f_0$ of 6.1 Hz, so it is a rather poor reference site

candidate, with not only a clear amplification peak reaching above 3 based on 162 recordings (top plot), but also exhibits directional sensitivity of 1.20 (bottom plot). In contrast, station IMMV appears to be a very good candidate, lacking any identifiable peak and having sensitivity around 1.07.

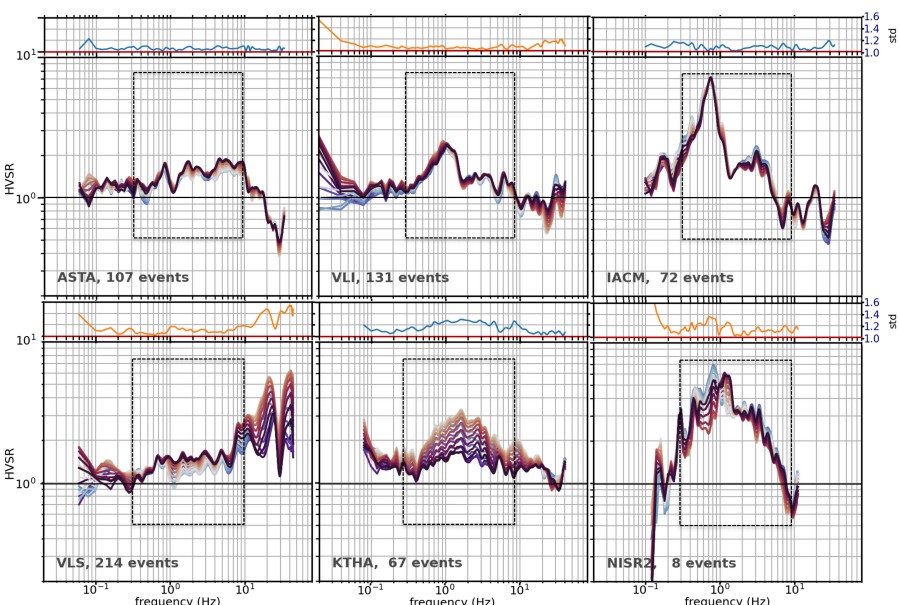

**Figure 6:** Indicative examples of HVSR: From left to right: low, medium and high amplification within the range of 0.3-10 Hz. From top to bottom, lower and higher variability with azimuth.

Figure 6 illustrates a few characteristic examples. Considered in the band 0.3-10 Hz, ASTA is the best reference candidate with no amplification and very low SD, followed by VLS, with rather higher variability (yet still a rather acceptable



reference below 10 Hz). VLI exhibits a weak but clear low-frequency resonance, while IACM a clear and very strong one, with also a rather clear first higher mode. None of these two show directional variability. KTHA and NISR2, on the other hand, show weak and strong peaks respectively which are rather broadband (not so 'peaky' as their counterparts VLI and IACM), and in addition possess a very high degree of directionality. The behaviour of most of these stations is certainly not what we would expect of 'rock stations'. Based on such geological 'labels', one might be typically consider them as reliable

reference stations, assuming no great amplification. Nonetheless, we see cases of either low or high-frequency (VLS) amplifications up to 6-8. In addition to that, for SD>1.20, what one would perceive as the 'reference' ground motion would depend very much on the orientation in which the sensor happened to be placed, since we see differences of up to factors of 2 or even 3 at certain frequencies. Figures S6 and S7 in the Supplement show results for all HH and HN stations respectively.

Based on such observations, we can group the stations of this study into a few indicative categories. We name these: reference station, small/large amplification, and small/large broadband amplification. Other groupings could be envisioned, but our goal here is to call attention to the main behaviours and how thy deviate from the expected (flat) rock response. Figure 7 shows these groups. We do not investigate on a station-by-station basis what exactly lies behind the amplification patterns we observe. Considering these are generally thought to be rock sites, we only mention a few possible interpretations

(other than geological misclassification). It is known that sharp high-frequency peaks can be due to shallow soft or weathered layers on bedrock, and their level will increase with the impedance (Vs) contrast between the two materials. A directional dependence of such a peak could signify 2D or 3D effects stemming from non-horizontal conditions. A low-frequency, relatively low peak could indicate a deep interface, likely between soft and harder rock. Given the hardness of the sites, another likely physical mechanism is topographic amplification, which would be expected to take place at specific

frequencies, depending on the overall material Vs and the height/width of the hill/slope/topographic feature (Geli et al., 1988; Ashford & Sitar, 1997). In this case, the spectral peak will also exhibit directionality, since such amplification is known to be strongest in a certain direction, such as transversely to the axis of a 2D ridge. We expect the interpretation to be more complex in the case of a 3D feature such as a hill or cave (instances of which exist in our database, see next section).

In Table 2 we compile some basic metadata for the 60 stations (such as $N_{rec}$, M range), along with the results of the

225 assessment presented in this section, namely: $f_0$, $A_0$, $f_1$, $S1$ (if applicable), $SD_{0.3-30}$, $SD_{1-10}$, $SD_{f0}$, and description of amplification. We also include an additional calculation: the maximum amplitude that the transfer function reaches if we correct the HVSR for the implicit amplification of the vertical component. To do so, we use the function proposed by Ito et al. (2020) called VACF (correction function for vertical amplification). This has its limitations, since VACF was calibrated on Japanese data, but we consider it a not illogical first approximation, coming from a region of similar (active) tectonic

regime. VACF has been constrained within a narrower band than used in this study, namely from 0.12-15 Hz. An example is shown in the Supplement (Fig. S3). Table 2 includes field $A_{0\_corr}$ in an indicative role, as a rough indication of the potential absolute amplification at the sites, and not to be used at face value for hazard or other calculations.


We note that any very strongly nonlinear recordings (though this is not very probable for rock/stiff conditions) would be eliminated at the visual inspection stage, while weaker ones may still remain, since we assume they would not bias the

235 ensemble mean results enough to merit a dedicated check. If present, we expect nonlinearity to decrease the level of high-frequency peaks. Since we are rather strict in our use of a threshold of 2 rather than 2.8 or 3, we believe it is not a grave issue.

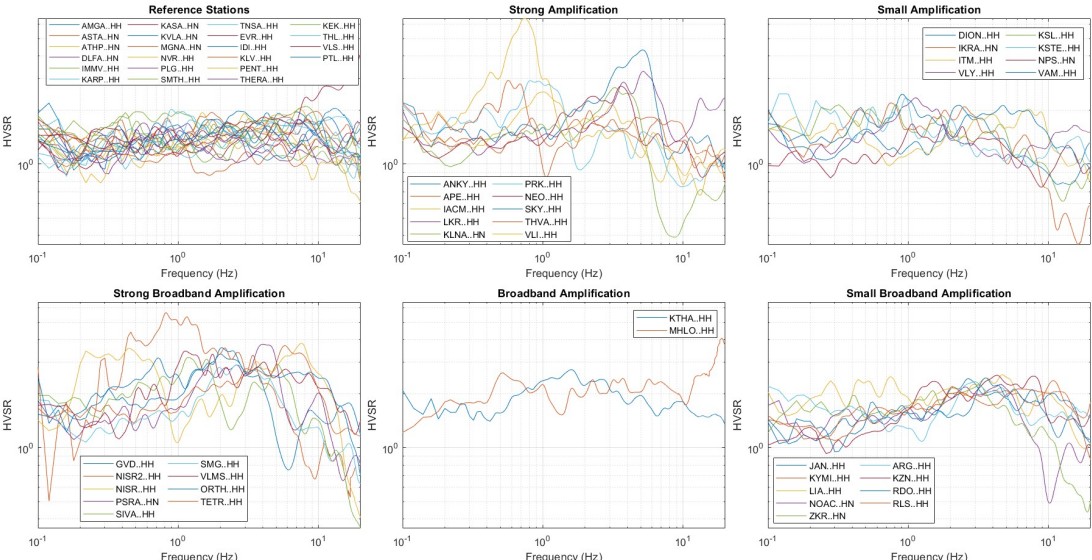

**Figure 7:** Groups of stations with similar amplification.

## 3 Compiling other station metadata

### 3.1 Rationale

There have been studies and projects dedicated to assessing the most useful parameters and proxies in describing site

conditions.

a. Cultrera et al. (2021) conducted a wide European survey including various end users and considering aspects such as cost and difficulty in procuring the parameters, which concluded that the preferred 7 indicators out of a total of 24 –some being admittedly not very common- are the following: 1. fundamental frequency $f_0$; 2. full Vs profile; 3. $Vs_{30}$; 4. depth to seismological bedrock; 5. depth to engineering bedrock; 6. surface geology; and 7. soil class. We note that some of these are

250 direct derivatives of others (3 hinging upon 2 and 7 depending on 2 and 4).





b. Lanzano et al. (2020) conducted a study in Central Italy focusing on rock sites in particular, and proposed an algorithm that takes into account 6 site descriptors, grading and combining them mathematically to produce an overall qualifier for characterising reference stations. Their proxies used to identify rock stations are: 1. housing/installation conditions; 2. topographic conditions; 3. surface geology (same as 6 above); 4. $Vs_{30}$ (same as 3 above); 5. shape of HVSR from noise or earthquakes (related to 1 above); 6. δs2s, the site-to-site term resulting from GMPE residuals analysis using response spectra, as an alternative estimate of the transfer function.

c. Pilz et al. (2020) assess reference stations at a European level from homogenized data considering the following parameters: 1. surface geology (as above); 2. slope/topography (as above); 3. HVSR (as above); 4. similarity of surface $κ_0$ (high-frequency site attenuation) to coda $κ_0$, which is considered as indicative of deeper conditions; 5. ML station residuals.

In the previous section we computed FAS-based HVSR for the first time for these stations, and in this section we compile all existing parameters we can find from various sources: housing/installation, topography/slope, surface geology, and $Vs_{30}$ (ad-hoc Vs profiles being almost non-existent in Greek seismic rock stations). To this literature-based collation, we also add insights based on site visits by NOA personnel. We believe this is important because geological maps constructed for an entire country inevitably contain errors and simplifications, whereas a site walkover of the station location by an experienced geologist provides additional reliability. Similarly, satellite-based estimates of slope/topography invariably include approximation, homogenisation and some lack of specificity depending on the size of the 'pixel', whereas again a site visit laves little doubt as to the exact nature of the landscape at the station.

### 3.2 Station installation

Table 3 compiles the information we found on housing and installation conditions at our 60 rock stations. Information for the HN stations is available from the website https://accelnet.gein.noa.gr/station-information/ (last accessed: December 2023), while additional detail specially for the HH stations is provided based on site visits, with more detailed descriptions given in the dedicated article on EIDA@NOA (Evangelidis et al., 2021). The last column of the table provides our assessment as to whether each station can be considered a reference station based on installation conditions. We note here that housing condition for HL network are vastly different to those of other countries, with explicit free-field conditions being rather rare. The Italian equivalent (Lanzano et al., 2020) only makes reference to two types of stations, free-field and in power towers, while the NOA network has had to make use of environments as diverse as monastery cells. However, in all cases where a 'vault' is mentioned, this is created within the structure hosting it by cutting around the station in a way so as to isolate its potential motion from that of the surrounding structure, hence avoiding soil-structure interaction effects.

### 3.3 Topography and slope

Table 4 compiles the information gathered on terrain slope and topographic conditions at our stations. There are various sources. For the HN stations, ESM (https://esm-db.eu/; Lanzano et al., 2021; Luzi et al., 2016) provide the slope in degrees along with their classification into four categories with the following code: T1: 'Flat surface, isolated slopes and cliffs with





average slope angle i≤15°'; T2: 'Slopes with average slope angle i>15°'; T3: 'Ridges with crest width significantly less than the base width and average slope angle 15°≤i≤30°'; T4: 'Ridges with crest width significantly less than the base width and

285 average slope angle i>30°'. For the HN stations again, Margaris et al. (2021) provide an estimate of slope which we have also converted into degrees and which for the most part almost coincides with the angles by ESM (save 2 stations marked in the table in bold italics, DLFA and NOAC, where however the difference does not cause a change in ESM code). For the entirety of stations studied here, additional detail is also provided based on site visits, where we group stations into the following categories: 1. Flat/shallow (<15) within 200 m; 2. Steep (<30) within 200 m; 3. Steep hill crest; 4. Near cliff. This

offers new information for about 35 stations for which no information was available before, some of them on various kinds of steep conditions.

### 3.4 Vs$_{30}$

Table 5 compiles the information gathered on Vs at our study's rock stations. There are again various sources. For the HN stations, ESM again provides the proxy-based Vs$_{30}$ using slope (and consequent EC8 soil class as per CEN, 2004), while

Margaris et al. (2021) provide a variety of estimates of Vs$_{30}$. A couple come from measurements in the vicinity of the stations (within 1 km, as per Stewart et al., 2014), while most are derived from proxies, using not only ground slope but also terrain, and a single value per station is given as preferred by that study. We note that although Stewart et al. (2014) was based on entire Vs profiles, that study did not release any profiles as functions of depth, but rather their derived average Vsz values over a given depth z. Finally, a couple of stations have been characterised ad hoc at the station location by NOA

within the national project HELPOS (Deliverable 2.5.3, Geophysical measurements at seismic stations). Between the three sources of information, namely ESM, Margaris et al. (2021) and HELPOS, there are in some cases discrepancies. The strongest contradictions that correspond to, say, a factor of 2-3 of difference in Vs$_{30}$ and a clear jump in site class, are marked in Table 5 in bold italics, such as ATHP, IACM, KASA, KSL, SMTH. In the case of measured Vs profiles on the spot (HELPOS), we consider those as the definitive Vs$_{30}$ estimates. However, in the case of measurement within 1 km distance

form the station, we believe their validity very much depends on lateral variations in stratigraphy and so do not attach more confidence to them than the proxy-based ones of ESM.

### 3.5 Geology

Table 6 compiles all the information gathered on surface geology at our study's rock stations. Information for the HN stations is available from the website https://accelnet.gein.noa.gr/station-information/ (last accessed: December 2023).

Description of the geological unit and age are provided for HN stations by Margaris et al. (2021). Finally, 17 of our 60 stations were also found in the list of Pilz et al. (2020) for European reference sites, and in those cases we also report the unified geological descriptors attributed by them according to the European Geological Data Infrastructure (EGDI). Two of those attributes were based on AI and are noted as such in the table.



One of the important features of this study is that we provide new information for the entirety of stations, consisting of
315 geological unit and age descriptions. This is based on the combination of site visit and walkover experience with the detailed
revisiting of maps and literature. The majority of stations were located in 53 geological maps (1:50,000 scale) published by
the Hellenic Survey of Geology and Mineral Exploration (HSGME) and their geology interpreted in conjunction with
knowledge of the local features from sit visits. Geological conditions for a couple of stations were derived from relevant
publications indicated in Table 6 with an asterisk. There are several contradictions between the various sources, too
numerous to discuss in detail here. Our best estimate after assessing all available information and experience is given in the
relevant columns 'this study'.

**Table 1.** General information and metadata for the stations in this study and statistics on the ground-motion data analysed.

| No. | Station code | Channel | Station Name | Network code | StLat (deg) | StLon (deg) | StEl (m) | Period | ML range | Nrec |
|---|---|---|---|---|---|---|---|---|---|---|
| 1 | AMGA | HNc | AMORGOS | HL | 36.83156 | 25.89384 | 308 | 2012-2019 | 4-6.2 | 60 |
| 2 | ANKY | HNc | ANTIKYTHIRA | HL | 35.86704 | 23.30117 | 143 | 2012-2021 | 4-6.6 | 110 |
| 3 | APE | HH | APEIRANTHOS, NAXOS | HL/GE | 37.07274 | 25.52301 | 608 | 2012-2021 | 4-6.3 | 162 |
| 4 | ARG | HH | ARCHANGELOS, RHODES | HL | 36.21356 | 28.12122 | 148 | 2012-2021 | 4-6.7 | 124 |
| 5 | ASTA | HN | ASTYPALAIA | HL | 36.54552 | 26.35295 | 64 | 2012-2020 | 4-6.7 | 129 |
| 6 | ATHP | HN | ATHENS-NEO PSYCHIKO | HL | 38.00080 | 23.77349 | 187 | 2020-2021 | 4-6.0 | 25 |
| 7 | DION | HNc | DIONYSOS ATTIKIS | HL | 38.07794 | 23.93306 | 460 | 2013-2016 | 4-6.3 | 33 |
| 8 | DLFA | HN | DELFOI | HL | 38.47836 | 22.49583 | 570 | 2012-2021 | 4-6.6 | 316 |
| 9 | EVR | HH | EVRITANIA | HT | 38.91657 | 21.81050 | 1037 | 2012-2021 | 4-6.1 | 296 |
| 10 | GVD | HNc | GAVDOS | HL | 34.83914 | 24.08738 | 170 | 2012-2017 | 4-6.2 | 86 |
| 11 | IACM | HNc | HERAKLEIO | HL | 35.30580 | 25.07090 | 45 | 2017-2021 | 4-6.3 | 72 |
| 12 | IDI | HH | ANOGEIA | HL/MN | 35.28878 | 24.89043 | 750 | 2012-2021 | 4-5.5 | 161 |
| 13 | IKRA | HN | AGIOS KIRIKOS IKARIA | HL | 37.61117 | 26.29283 | 30 | 2012-2017 | 4-6.3 | 79 |
| 14 | IMMV | HNc | CHANIA, CRETE | HL/GE | 35.46060 | 23.98110 | 230 | 2012-2021 | 4-6.2 | 142 |
| 15 | ITM | HNc | ITHOMI MESSINIA | HL | 37.17872 | 21.92522 | 423 | 2018-2017 | 4-6.0 | 217 |
| 16 | JAN | HNc | IOANNINA | HL | 39.65616 | 20.84874 | 526 | 2012-2022 | 4-6.6 | 177 |
| 17 | KARP | HNc | KARPATHOS | HL | 35.54710 | 27.16106 | 524 | 2012-2021 | 4-6.7 | 284 |
| 18 | KASA | HN | KASSIOPI | HL | 39.74628 | 19.93542 | 65 | 2012-2018 | 4-6.0 | 73 |
| 19 | KEK | HH | KERKYRA | HL/MN | 39.71270 | 19.79623 | 227 | 2012-2022 | 4-6.6 | 98 |
| 20 | KLNA | HN | KALYMNOS | HL | 36.95708 | 26.97274 | 28 | 2013-2021 | 4-6.7 | 240 |
| 21 | KLV | HH | KALAVRITA | HL | 38.04350 | 22.15040 | 758 | 2012-2021 | 4-6.6 | 322 |
| 22 | KSL | HNc | KASTELLORIZO | HL | 36.15031 | 29.58561 | 64 | 2012-2021 | 4-6.7 | 77 |
| 23 | KSTE | HNc | KASTELLI, CRETE | HL | 35.18010 | 25.33720 | 395 | 2021-2021 | 3.5-5.7 | 19 |
| 24 | KTHA | HNc | KYTHIRA | HL | 36.25660 | 23.06210 | 360 | 2013-2021 | 4-6.2 | 67 |
| 25 | KVLA | HN | KAVALA | HL | 40.93704 | 24.38591 | 122 | 2012-2022 | 4-6.1 | 52 |
| 26 | KYMI | HNc | KYMI | HL | 38.63315 | 24.10014 | 259 | 2014-2021 | 4-6.7 | 99 |
| 27 | KZN | HH | KOZANI | HL | 40.30331 | 21.78209 | 791 | 2012-2021 | 4-5.9 | 121 |
| 28 | LIA | HNc | LIMNOS | HL | 39.89725 | 25.18055 | 67 | 2012-2022 | 4-6.7 | 107 |
| 29 | LKR | HH | ATALANTI LOKRIDA | HL | 38.64957 | 22.99881 | 192 | 2012-2017 | 4-6.3 | 94 |
| 30 | MGNA | HN | MEGANISSI LEUKADA | HL | 38.65606 | 20.79116 | 58 | 2012-2014 | 4-5.8 | 75 |
| 31 | MHLO | HH | PLAKA, MILOS ISLAND | HL | 36.68984 | 24.40171 | 175 | 2012-2021 | 4-6.3 | 190 |
| 32 | NEO | HH | NEOCHORI VOLOS | HL/MN | 39.30567 | 23.22189 | 510 | 2012-2022 | 4-6.3 | 141 |
| 33 | NISR | HNc | NISYROS ISLAND | HL | 36.61060 | 27.13090 | 44 | 2021-2021 | 4-6.7 | 94 |
| 34 | NISR2 | HH | VOLCANOGOLY MUSEUM, NISYROS | HL | 36.57441 | 27.17666 | 423 | 2021-2021 | 4-5.7 | **8** |
| 35 | NOAC | HNc | ATHENS- THISSEIO | HL | 37.97384 | 23.71767 | 93 | 2012-2018 | 4-6.3 | 128 |
| 36 | NPS | HH | NEAPOLIS CRETE | HL | 35.26134 | 25.61037 | 288 | 2012-2016 | 4-6.2 | 58 |
| 37 | NVR | HNc | KATO NEVROKOPI | HL | 41.34846 | 23.86517 | 627 | 2012-2021 | 4-6.3 | 48 |
| 38 | ORTH | HH | ORTHONIES, ZAKYNTHOS | HL | 37.85112 | 20.69627 | 450 | 2018-2020 | 4-5.9 | 65 |
| 39 | PENT | HH | PENTALOFOS KOZANIS | HL | 40.19588 | 21.13842 | 1096 | 2012-2021 | 4-6.0 | 55 |
| 40 | PLG | HNc | POLIGIROS CHALKIDIKI | HL | 40.37328 | 23.44443 | 566 | 2013-2013 | 4-6.3 | 42 |
| 41 | PRK | HNc | AGIA PARASKEVI LESVOS | HL | 39.24565 | 26.26499 | 130 | 2013-2022 | 4-6.7 | 100 |

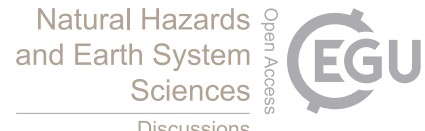
| 42 | PSRA | HN | PSARA | HL | 38.53978 | 25.56202 | 13 | 2012-2018 | 4-6.3 | 71 |
| 43 | PTL | HH | PENTELI | HL | 38.04730 | 23.86380 | 500 | 2012-2021 | 4-6.6 | 142 |
| 44 | RDO | HH | RODOPI | HL | 41.14503 | 25.53553 | 116 | 2012-2020 | 4-6.1 | 58 |
| 45 | RLS | HH | RIOLOS KATO ACHAEA | HL | 38.05586 | 21.46475 | 97 | 2012-2021 | 4-6.6 | 328 |
| 46 | SIVA | HNc | SIVAS CRETE | HL/GE | 35.01777 | 24.81204 | 96 | 2012-2021 | 4-6.3 | 42 |
| 47 | SKY | HH | SKYROS | HL | 38.88310 | 24.54820 | 268 | 2012-2022 | 4-6.3 | 70 |
| 48 | SMG | HNc | SAMOS | HL | 37.70425 | 26.83772 | 348 | 2020-2022 | 4-6.3 | 51 |
| 49 | SMTH | HNc | SAMOTHRAKI | HL | 40.47094 | 25.53045 | 365 | 2012-2022 | 4-6.7 | 68 |
| 50 | TETR | HH | TETRAKOMO | HL | 39.34450 | 21.27467 | 942 | 2018-2022 | 4-5.9 | 66 |
| 51 | THERA | HH | ANCIENT THERA, SANTORINI | HL/GE | 36.36699 | 25.47526 | 288 | 2019-2021 | 4-6.3 | 83 |
| 52 | THL | HH | KLOKOTOS | HL/MN | 39.56468 | 22.01440 | 86 | 2012-2021 | 4-5.4 | 116 |
| 53 | THVA | HH | IEK THIVAS | HL | 38.32983 | 23.33601 | 214 | 2020-2021 | 4-5.9 | 27 |
| 54 | TNSA | HN | TINOS | HL | 37.53942 | 25.16310 | 21 | 2012-2021 | 4-6.7 | 125 |
| 55 | VAM | HH | VAMOS | HL | 35.40700 | 24.19970 | 225 | 2012-2021 | 4-6.3 | 139 |
| 56 | VLI | HH | VELIES LAKONIA | HL | 36.71803 | 22.94686 | 220 | 2012-2021 | 4-6.2 | 131 |
| 57 | VLMS | HNc | VOLIMES- ZAKYNTHOS | HL | 37.87670 | 20.66293 | 431 | 2014-2015 | 4-5.8 | 56 |
| 58 | VLS | HH | VALSAMATA KEFALONIA | HL | 38.17683 | 20.58860 | 402 | 2012-2021 | 4-5.9 | 214 |
| 59 | VLY | HH | BOYLA ATTIKHS | HL | 37.85240 | 23.79420 | 256 | 2012-2022 | 4-6.3 | 103 |
| 60 | ZKR | HNc | ZAKROS | HL/GE | 35.11470 | 26.21691 | 254 | 2012-2019 | 4-6.2 | 106 |

**Table 2.** Detailed results of the HVSR analysis for the stations in this study.

| No. | Station code | $f_0$ (Hz) | $A_0$ | $f_1$ (Hz) | $A_1$ | $A_{0\_corr}$ | $SD_{0.3-30}$ | $SD_{1-10}$ | $SD_{f0}$ | Amplification group | Potential reference site? |
|---|---|---|---|---|---|---|---|---|---|---|---|
| 1 | AMGA | - | - | - | - | - | 1.05 | 1.04 | | Reference station | yes |
| 2 | ANKY | 0.8 | 2.75 | - | - | 5 | 1.11 | 1.06 | 1.36 | Strong LF amplification | no |
| 3 | APE | 6.2 | 3.4 | 20 | 3 | 7 | 1.19 | 1.09 | 1.20 | Strong HF amplification | no |
| 4 | ARG | 3 | 2.3 | - | - | 5 | 1.18 | 1.13 | 1.21 | Small broadband amplification | ok |
| 5 | ASTA | - | - | - | - | | 1.05 | 1.07 | - | Reference station | yes |
| 6 | ATHP | - | - | - | - | - | 1.12 | 1.1 | - | Reference station | yes |
| 7 | DION | 2.3 | 2.1 | - | - | 4.2 | 1.13 | 1.13 | 1.09 | Small amplification | ok |
| 8 | DLFA | - | - | - | - | - | 1.07 | 1.05 | - | Reference station | yes |
| 9 | EVR | - | - | - | - | - | 1.13 | 1.07 | - | Reference station | yes |
| 10 | GVD | 0.3 | 2.1 | 2 | 3.3 | 5.5 | 1.06 | 1.05 | 1.06 | Broadband strong amplification | no |
| 11 | IACM | 0.75 | 6.5 | 3.1 | 2.1 | 11 | 1.08 | 1.07 | 1.05 | Very strong LF amplification | no |
| 12 | IDI | - | - | - | - | 2.6 | 1.09 | 1.06 | | Reference station | yes |
| 13 | IKRA | 0.4 | 2.1 | 0.75 | 2.3 | 1.9 | 1.1 | 1.07 | 1.09 | Small amplification | ok |
| 14 | IMMV | - | - | - | - | - | 1.06 | 1.07 | | Reference station | yes |
| 15 | ITM | - | - | - | - | | 1.11 | 1.09 | | Small HF amplification | ok |
| 16 | JAN | - | - | - | - | 4.1 | 1.1 | 1.07 | 1.02 | Small broadband amplification | ok |
| 17 | KARP | - | - | - | - | 3.9 | 1.11 | 1.04 | 1.05 | Reference station | yes |
| 18 | KASA | - | - | - | - | | 1.07 | 1.07 | | Reference station | yes |
| 19 | KEK | - | - | - | - | 5 | 1.17 | 1.17 | 1.23 | Small HF amplification | ok |
| 20 | KLNA | 3.2 | 2.7 | - | - | | 1.09 | 1.11 | 1.19 | Strong amplification | no |
| 21 | KLV | - | - | - | - | | 1.06 | 1.06 | | Reference Station | yes |
| 22 | KSL | - | - | - | - | | 1.09 | 1.17 | | Small LF amplification | ok |
| 23 | KSTE | - | - | - | - | 4.7 | 1.13 | 1.13 | 1.06 | Small LF amplification | ok |
| 24 | KTHA | 1.6 | 2.7 | - | - | 3.6 | 1.11 | 1.2 | 1.30 | Broadband amplification | no |
| 25 | KVLA | - | - | - | - | | 1.07 | 1.04 | | Reference Station | yes |
| 26 | KYMI | 11 | 2.4 | - | - | 4.8 | 1.16 | 1.13 | 1.25 | Small broadband amplification | ok |
| 27 | KZN | 1.5 | 2.3 | 4.5 | 2.5 | 3.7 | 1.13 | 1.14 | 1.15 | Small broadband amplification | ok |
| 28 | LIA | 25 | 3.2 | - | - | 5.1 | 1.11 | 1.1 | 1.11 | Small broadband amplification | ok |
| 29 | LKR | 3.8 | 2.9 | 5.2 | 3.3 | 7.1 | 1.23 | 1.13 | 1.14 | Strong Amplification | no |
| 30 | MGNA | - | - | - | - | | 1.1 | 1.09 | | Reference Station | yes |
| 31 | MHLO | 0.5 | 2.6 | - | - | 3.1 | 1.1 | 1.07 | 1.11 | Broadband amplification | no |
| 32 | NEO | 3.6 | 2.7 | - | - | 5 | 1.14 | 1.13 | 1.11 | Strong amplification | no |
| 33 | NISR | 7.7 | 3.8 | - | - | 8.5 | 1.14 | 1.2 | 1.09 | Broadband strong amplification | no |
| 34 | NISR2 | 0.8 | 5.6 | - | - | 10 | 1.15 | 1.12 | 1.33 | Broadband strong amplification | no |
| 35 | NOAC | - | - | - | - | | 1.11 | 1.14 | | Small broadband amplification | ok |
| 36 | NPS | - | - | - | - | 4.1 | 1.1 | 1.09 | 1.12 | Reference Station | yes |
| 37 | NVR | 7.3 | 2 | - | - | 4 | 1.09 | 1.05 | 1.05 | Small HF amplification | ok |
| 38 | ORTH | 0.7 | 3.2 | 2 | 3.6 | 4.3 | 1.07 | 1.1 | 1.10 | Broadband strong amplification | no |
| 39 | PENT | - | - | - | - | | 1.13 | 1.08 | | Reference Station | yes |
| 40 | PLG | - | - | - | - | | 1.09 | 1.04 | | Reference Station | yes |
| 41 | PRK | 1.1 | 2.9 | - | - | 6 | 1.1 | 1.07 | 1.06 | Strong LF amplification | no |
| 42 | PSRA | 4 | 3.8 | - | - | 8.5 | 1.05 | 1.06 | 1.03 | Broadband strong amplification | no |





| No. | Station | | | | | | | | | | |
|-----|---------|---|---|---|---|---|------|------|------|------|------|
| 43 | PTL | - | - | - | - | | 1.07 | 1.08 | | Reference Station | yes |
| 44 | RDO | 3.5 | 2.4 | 27 | 2.6 | 3.6 | 1.13 | 1.06 | 1.07 | Small broadband amplification | ok |
| 45 | RLS | - | - | - | - | | 1.14 | 1.06 | | Small broadband amplification | ok |
| 46 | SIVA | 1.2 | 3.2 | 5 | 2.9 | 5.8 | 1.12 | 1.08 | 1.04 | Broadband strong amplification | no |
| 47 | SKY | 5.2 | 4.3 | - | - | 10 | 1.11 | 1.07 | 1.15 | Very strong amplification | no |
| 48 | SMG | 2 | 2.8 | 2.7 | 2.9 | 5 | 1.15 | 1.09 | 1.13 | Broadband strong amplification | no |
| 49 | SMTH | - | - | - | - | | 1.05 | 1.09 | | Reference station | yes |
| 50 | TETR | 5.7 | 3.7 | - | - | 7.3 | 1.18 | 1.13 | 1.09 | Broadband strong amplification | no |
| 51 | THERA | - | - | - | - | | 1.23 | 1.21 | | Reference Station | yes |
| 52 | THL | - | - | - | - | | 1.05 | 1.06 | | Reference Station | yes |
| 53 | THVA | 0.6 | 2.9 | - | - | 3.7 | 1.15 | 1.14 | 1.12 | Strong amplification | no |
| 54 | TNSA | - | - | - | - | | 1.08 | 1.12 | | Reference station | yes |
| 55 | VAM | 0.45 | 2.1 | 0.9 | 2.45 | 2.1 | 1.12 | 1.17 | 1.21 | Small amplification | ok |
| 56 | VLI | 1 | 2.52 | - | - | 5 | 1.09 | 1.06 | 1.04 | Strong amplification | no |
| 57 | VLMS | 1.1 | 3.5 | 1.8 | 3.3 | 7 | 1.13 | 1.11 | 1.24 | Broadband strong amplification | no |
| 58 | VLS | 21 | 4.4 | - | - | - | 1.24 | 1.12 | 1.45 | Reference station | yes |
| 59 | VLY | 1.1 | 2.3 | - | - | 4.4 | 1.08 | 1.1 | 1.05 | Small amplification | ok |
| 60 | ZKR | 3.1 | 2.2 | 5 | - | - | 1.12 | 1.09 | 1.09 | Small broadband amplification | no |

**Table 3.** Housing and installation conditions for the stations in this study.

| No. | Station code | StEl (m) | Building type (from accelnet) | Installation condition - site visit | Potential reference site? |
|-----|--------------|----------|-------------------------------|-------------------------------------|---------------------------|
| 1 | AMGA | 308 | 1-floor RC | not free field | no |
| 2 | ANKY | 143 | - | free field only for HH | no |
| 3 | APE | 608 | - | vault in building | yes |
| 4 | ARG | 148 | - | vault in building | yes |
| 5 | ASTA | 64 | 2-floor adobe masonry | not free field | no |
| 6 | ATHP | 187 | 3-floor RC | not free field | no |
| 7 | DION | 460 | - | cave | no |
| 8 | DLFA | 570 | 3-floor RC | not free field | no |
| 9 | EVR | 1037 | - | vault in building | yes |
| 10 | GVD | 170 | - | free field | yes |
| 11 | IACM | 45 | - | free field | yes |
| 12 | IDI | 750 | - | underground vault | yes |
| 13 | IKRA | 30 | 2-floor RC | not free field | no |
| 14 | IMMV | 230 | - | monastery cell on rock | no |
| 15 | ITM | 423 | - | vault in building | yes |
| 16 | JAN | 526 | 1-floor RC | vault in building | yes |
| 17 | KARP | 524 | - | vault in small building | yes |
| 18 | KASA | 65 | 2-floor RC | not free field | no |
| 19 | KEK | 227 | - | vault in small building | yes |
| 20 | KLNA | 28 | 3-floor RC | not free field | no |
| 21 | KLV | 758 | - | digged cave | no |
| 22 | KSL | 64 | - | vault in small building | yes |
| 23 | KSTE | 395 | - | vault in small building | yes |
| 24 | KTHA | 360 | - | monastery cell on rock | no |
| 25 | KVLA | 122 | 2-floor RC | not free field | no |
| 26 | KYMI | 259 | 1-floor RC | vault in small building | yes |
| 27 | KZN | 791 | 1-floor RC | vault in building | yes |
| 28 | LIA | 67 | 1-st RC | free field only for HH | no |
| 29 | LKR | 192 | 1-floor RC | vault in building | yes |
| 30 | MGNA | 58 | 2-floor RC | not free field | no |
| 31 | MHLO | 175 | - | not free field | no |
| 32 | NEO | 510 | - | vault in building | yes |
| 33 | NISR | 44 | - | not free field | no |
| 34 | NISR2 | 423 | - | free field | yes |
| 35 | NOAC | 93 | - | vault in building | yes |
| 36 | NPS | 288 | - | vault in building | yes |
| 37 | NVR | 627 | 1-floor RC | free field | yes |
| 38 | ORTH | 450 | - | not free field | no |
| 39 | PENT | 1096 | - | not free field | no |
| 40 | PLG | 566 | 1-floor RC | vault in building | yes |





| 41 | PRK | 130 | 1-floor RC | vault in building | yes |
|----|-----|-----|------------|-------------------|-----|
| 42 | PSRA | 13 | 2-floor RC | not free field | no |
| 43 | PTL | 500 | - | vault in building | yes |
| 44 | RDO | 116 | - | vault in small building | yes |
| 45 | RLS | 97 | - | vault in building | yes |
| 46 | SIVA | 96 | 1-floor masonry | free field | yes |
| 47 | SKY | 268 | - | not free field | no |
| 48 | SMG | 348 | - | vault in small building | yes |
| 49 | SMTH | 365 | 1-floor RC | underground vault | yes |
| 50 | TETR | 942 | - | vault in small building | yes |
| 51 | THERA | 288 | - | free field | yes |
| 52 | THL | 86 | 1-floor RC | dug vault in rock | yes |
| 53 | THVA | 214 | 1-floor RC | not free field | no |
| 54 | TNSA | 21 | 2-floor RC | not free field | no |
| 55 | VAM | 225 | - | vault in building | yes |
| 56 | VLI | 220 | - | vault in small building | yes |
| 57 | VLMS | 431 | free-field | free field | yes |
| 58 | VLS | 402 | - | vault in building | yes |
| 59 | VLY | 256 | - | free field | yes |
| 60 | ZKR | 254 | 1-floor RC | not free field | no |

**Table 4.** Topography and slope conditions for the stations in this study.

| No. | Station code | StEl (m) | Topography code (ESM) | Slope angle° (ESM) | Slope (Marg2021) | Slope angle° based on Marg2021 | Topography assessment by site visits | Potential reference site? |
|-----|--------------|----------|----------------------|--------------------|-------------------|-------------------------------|--------------------------------------|--------------------------|
| 1 | AMGA | 308 | T1 | 3 | 0.047 | 3 | Flat/shallow (<15) within 200 m | yes |
| 2 | ANKY | 143 | T1 | 8 | 0.138 | 8 | Flat/shallow (<15) within 200 m | yes |
| 3 | APE | 608 | - | - | - | - | Flat/shallow (<15) within 200 m | yes |
| 4 | ARG | 148 | T1 | - | - | - | Flat/shallow (<15) within 200 m | yes |
| 5 | ASTA | 64 | T1 | 9 | 0.14 | 8 | Flat/shallow (<15) within 200 m | yes |
| 6 | ATHP | 187 | T1 | 2 | - | - | Flat/shallow (<15) within 200 m | yes |
| 7 | DION | 460 | T3 | 2 | - | - | Flat/shallow (<15) within 200 m | yes |
| 8 | DLFA | 570 | T4 | *38* | 0.502 | *27* | Steep (<30) within 200 m | no |
| 9 | EVR | 1037 | - | - | - | - | Flat/shallow (<15) within 200 m | yes |
| 10 | GVD | 170 | T1 | 2 | 0.035 | 2 | Flat/shallow (<15) within 200 m | yes |
| 11 | IACM | 45 | T1 | 11 | - | - | Flat/shallow (<15) within 200 m | yes |
| 12 | IDI | 750 | - | - | - | - | Steep (<30) within 200 m | no |
| 13 | IKRA | 30 | - | - | 0.129 | 7 | Flat/shallow (<15) within 200 m | yes |
| 14 | IMMV | 230 | T2 | 18 | - | - | Flat/shallow (<15) within 200 m | yes |
| 15 | ITM | 423 | T1 | 13 | 0.223 | 13 | Flat/shallow (<15) within 200 m | yes |
| 16 | JAN | 526 | T1 | 2 | 0.056 | 3 | Flat/shallow (<15) within 200 m | yes |
| 17 | KARP | 524 | T3 | 11 | - | - | Steep hill crest | no |
| 18 | KASA | 65 | T4 | 17 | 0.321 | 18 | Steep (<30) within 200 m | no |
| 19 | KEK | 227 | - | - | - | - | Steep (<30) within 200 m | no |
| 20 | KLNA | 28 | T1 | 4 | 0.065 | 4 | Flat/shallow (<15) within 200 m | yes |
| 21 | KLV | 758 | - | - | - | - | Steep (<30) within 200 m | no |
| 22 | KSL | 64 | T3 | 9 | 0 | 0 | Steep hill crest | no |
| 23 | KSTE | 395 | - | - | - | - | Steep (<30) within 200 m | no |
| 24 | KTHA | 360 | T4 | 7 | - | - | Steep hill crest | no |
| 25 | KVLA | 122 | T3 | 9 | 0.122 | 7 | Flat/shallow (<15) within 200 m | yes |
| 26 | KYMI | 259 | T2 | 11 | - | - | Steep hill crest | no |
| 27 | KZN | 791 | - | - | 0.113 | 6 | Flat/shallow (<15) within 200 m | yes |
| 28 | LIA | 67 | T1 | 3 | 0.073 | 4 | Flat/shallow (<15) within 200 m | yes |
| 29 | LKR | 192 | - | - | - | - | Flat/shallow (<15) within 200 m | yes |
| 30 | MGNA | 58 | T1 | 5 | 0.07 | 4 | Flat/shallow (<15) within 200 m | yes |
| 31 | MHLO | 175 | - | - | - | - | Flat/shallow (<15) within 200 m | yes |
| 32 | NEO | 510 | - | - | - | - | Flat/shallow (<15) within 200 m | yes |
| 33 | NISR | 44 | - | - | - | - | Near cliff | no |
| 34 | NISR2 | 423 | - | - | - | - | Near cliff | no |
| 35 | NOAC | 93 | T1 | *7* | 0.225 | *13* | Flat/shallow (<15) within 200 m | yes |
| 36 | NPS | 288 | - | - | - | - | Flat/shallow (<15) within 200 m | yes |
| 37 | NVR | 627 | T1 | 15 | 0.259 | 15 | Flat/shallow (<15) within 200 m | yes |
| 38 | ORTH | 450 | - | - | - | - | Flat/shallow (<15) within 200 m | yes |





| No. | Station | Vs30 | EC8 | | | | | | |
|---|---|---|---|---|---|---|---|---|---|
| 39 | PENT | 1096 | - | - | - | - | Steep (<30) within 200 m | no |
| 40 | PLG | 566 | T1 | 5 | 0.073 | 4 | Flat/shallow (<15) within 200 m | yes |
| 41 | PRK | 130 | T1 | 5 | - | - | Flat/shallow (<15) within 200 m | yes |
| 42 | PSRA | 13 | T1 | 2 | 0.047 | 3 | Flat/shallow (<15) within 200 m | yes |
| 43 | PTL | 500 | - | - | - | - | Flat/shallow (<15) within 200 m | yes |
| 44 | RDO | 116 | - | - | - | - | Flat/shallow (<15) within 200 m | yes |
| 45 | RLS | 97 | - | - | - | - | Flat/shallow (<15) within 200 m | yes |
| 46 | SIVA | 96 | T1 | 7 | 0.007 | 0 | Flat/shallow (<15) within 200 m | yes |
| 47 | SKY | 268 | - | - | - | - | Steep (<30) within 200 m | no |
| 48 | SMG | 348 | T3 | 2 | - | - | Flat/shallow (<15) within 200 m | yes |
| 49 | SMTH | 365 | T2 | 23 | 0.313 | 17 | Steep (<30) within 200 m | no |
| 50 | TETR | 942 | - | - | - | - | Steep (<30) within 200 m | no |
| 51 | THERA | 288 | - | - | - | - | Steep (<30) within 200 m | no |
| 52 | THL | 86 | - | - | - | - | Flat/shallow (<15) within 200 m | yes |
| 53 | THVA | 214 | - | - | - | - | Flat/shallow (<15) within 200 m | yes |
| 54 | TNSA | 21 | T1 | 2 | 0.054 | 3 | Flat/shallow (<15) within 200 m | yes |
| 55 | VAM | 225 | - | - | - | - | Flat/shallow (<15) within 200 m | yes |
| 56 | VLI | 220 | - | - | - | - | Flat/shallow (<15) within 200 m | yes |
| 57 | VLMS | 431 | T1 | 4 | 0.082 | 5 | Flat/shallow (<15) within 200 m | yes |
| 58 | VLS | 402 | - | - | 0.048 | 3 | Flat/shallow (<15) within 200 m | yes |
| 59 | VLY | 256 | - | - | - | - | Flat/shallow (<15) within 200 m | yes |
| 60 | ZKR | 254 | T1 | 3 | 0.114 | 6 | Flat/shallow (<15) within 200 m | yes |

**Table 5.** Vs30 estimates for the stations in this study.

| No. | Station code | EC8 from ESM | Vs30 from ESM (m/s) | Measured Vs30 (Marg2021) (m/s) | Profile SiteCode (Stew2014) | Vs30 from geology/slope proxy (Marg2021) | Vs30 from terrain proxy (Marg2021) | Preferred Vs30 (Marg2021) | Measured Vs30 (HELPOS) | Potential reference site? |
|---|---|---|---|---|---|---|---|---|---|---|
| 1 | AMGA | B | 502 | - | - | 589 | 475 | 529 | | no |
| 2 | ANKY | B | 760 | - | - | 589 | 475 | 529 | | yes - ESM |
| 3 | APE | - | - | - | - | - | - | - | | - |
| 4 | ARG | B | 479 | - | - | - | - | - | - | no |
| 5 | ASTA | B | 793 | - | - | 538 | 475 | 506 | | yes - ESM |
| 6 | ATHP | B | *453* | - | - | - | - | - | *975* | yes - Helpos |
| 7 | DION | B | 459 | - | - | - | - | - | | no |
| 8 | DLFA | A | 1666 | - | - | -999 | 475 | 475 | | yes - ESM |
| 9 | EVR | - | - | - | - | - | - | - | | - |
| 10 | GVD | B | 411 | - | - | 589 | 475 | 529 | | no |
| 11 | IACM | A | *844* | - | - | - | - | - | *273* | no |
| 12 | IDI | - | - | - | - | - | - | - | | - |
| 13 | IKRA | - | - | - | - | 589 | 475 | 529 | | no |
| 14 | IMMV | A | 1036 | - | - | - | - | - | | yes - ESM |
| 15 | ITM | A | 882 | - | - | 610 | 475 | 538 | | yes - ESM |
| 16 | JAN | B | 399 | - | - | 589 | 365 | 464 | | no |
| 17 | KARP | A | 835 | - | - | - | - | - | | yes - ESM |
| 18 | KASA | A | *1006* | - | - | 589 | 475 | *529* | | yes - ESM |
| 19 | KEK | - | - | - | - | - | - | - | | - |
| 20 | KLNA | B | 530 | - | - | 371 | 365 | 368 | | no |
| 21 | KLV | - | - | - | - | - | - | - | | - |
| 22 | KSL | B | *781* | - | - | 137 | - | *137* | | yes - ESM |
| 23 | KSTE | - | - | - | - | - | - | - | | - |
| 24 | KTHA | B | 692 | - | - | - | - | - | | no |
| 25 | KVLA | B | 782 | - | - | 528 | 475 | 501 | | yes - ESM |
| 26 | KYMI | A | 844 | - | - | - | - | - | | yes - ESM |
| 27 | KZN | - | - | - | - | 589 | 475 | 529 | | no |
| 28 | LIA | B | 471 | - | - | 589 | 475 | 529 | | no |
| 29 | LKR | - | - | - | - | - | - | - | | - |



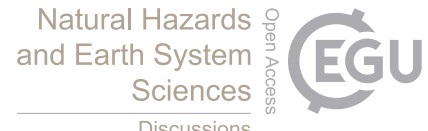
| 30 | MGNA | B | 595 | - | - | 589 | 365 | 464 | no |
|---|---|---|---|---|---|---|---|---|---|
| 31 | MHLO | - | - | - | - | - | - | - | - |
| 32 | NEO | - | - | - | - | - | - | - | - |
| 33 | NISR | - | - | - | - | - | - | - | - |
| 34 | NISR2 | - | - | - | - | - | - | - | - |
| 35 | NOAC | B | 719 | - | - | 589 | 475 | 529 | no |
| 36 | NPS | - | - | - | - | - | - | - | - |
| 37 | NVR | A | 955 | - | - | 556 | 475 | 514 | yes - ESM |
| 38 | ORTH | - | - | - | - | - | - | - | - |
| 39 | PENT | - | - | - | - | - | - | - | - |
| 40 | PLG | B | 608 | - | - | 492 | 519 | 506 | no |
| 41 | PRK | B | 604 | - | - | - | - | - | no |
| 42 | PSRA | B | 412 | - | - | 589 | 497 | 541 | no |
| 43 | PTL | - | - | - | - | - | - | - | - |
| 44 | RDO | - | - | - | - | - | - | - | - |
| 45 | RLS | - | - | - | - | - | - | - | - |
| 46 | SIVA | B | 706 | 492 | - | 353 | 365 | 492 | no |
| 47 | SKY | - | - | - | - | - | - | - | - |
| 48 | SMG | B | 451 | - | - | - | - | - | no |
| 49 | SMTH | A | *1167* | - | - | 589 | 475 | *529* | yes - ESM |
| 50 | TETR | - | - | - | - | - | - | - | - |
| 51 | THERA | - | - | - | - | - | - | - | - |
| 52 | THL | - | - | - | - | - | - | - | - |
| 53 | THVA | - | - | - | - | - | - | - | - |
| 54 | TNSA | B | 401 | - | - | 589 | 475 | 529 | no |
| 55 | VAM | - | - | - | - | - | - | - | - |
| 56 | VLI | - | - | - | - | - | - | - | - |
| 57 | VLMS | B | 527 | - | - | 589 | 519 | 553 | no |
| 58 | VLS | - | - | 872 | - | 461 | 365 | 872 | yes - Marg21 |
| 59 | VLY | - | - | - | - | - | - | - | - |
| 60 | ZKR | B | 522 | 877 | - | 589 | 519 | 877 | yes - Marg21 |

**Table 6.** Geological conditions for the stations in this study.

| No. | Station code | Geologic description - accelnet | Description (Marg2021) | Geologic Age (Marg2021) | Reference site as per Pilz et al. (2020) from EGDI | Geologic unit - this study | Geologic age - this study | Reference used - this study (HSGME map sheet or other*) | Potential reference site? |
|---|---|---|---|---|---|---|---|---|---|
| 1 | AMGA | Schist | Alluvial deposits | Triassic-Jurassic | - | Upper schist group (Paleogene flysch) | Middle-Upper Eocene-Oligocene | Amorgos-Donoussa Islands Sheet | yes |
| 2 | ANKY | | Alluvial deposits | Upper Cretaceous | claystone | Marls, sandstones, conglomerates | Neogene (Lower Tortonian-Upper Pliocene) | Antikithira Island Sheet | likely |
| 3 | APE | | - | - | Felsic rock | Metamorphic Complex | Pre-permian | Naxos Island Sheet | yes |
| 4 | ARG | | | | - | Sgourou Formation: Marls with sand and and gravel | Upper Pliocene to Pleistocene | North Rhodes | no |
| 5 | ASTA | Flysch | Alluvial deposits | Upper Eocene - Oligocene | - | Gavrovo-Tripolis Zone: Flysch mainly sandstones and conglomerates | Upper Eocene-Oligocene (?) | Astipalaia Island Sheet | no |
| 6 | ATHP | 0-2m fill material/2-6m poorly cemented conglomerate/6 | - | - | - | Alpine formations between the autochonous Almyropotamos -Attiki Unit and the Pelagonian | Upper Cretaceous | Kifissia Sheet | no |



| # | Code | | | | | | | | |
|---|------|---|---|---|---|---|---|---|---|
| | | -20m very weathered sandstone/20-30m sandstone | - | - | | zone (Afidnae-Tourkovounia Unit) | | | |
| 7 | DION | | - | - | metamorphic rock (AI) | Alpine formations between the autochonous Almyropotamos -Attiki Unit and the Pelagonian zone (Almyropotamos-Attiki autochthonous unit) | Middle Eocene | Kifissia Sheet | yes |
| 8 | DLFA | Limestone | Alluvial deposits | Quaternary | claystone | Parnassos-Ghiona Series (limestones) | Undivined Upper Cretaceous | Delfi Sheet | yes |
| 9 | EVR | | - | - | sandstone | Pindos series (pelagic limestones) | Maestrichtian-Danian | Karpenision Sheet | yes |
| 10 | GVD | | Alluvial deposits | Jurassic - Cretaceous | - | Clays and Marls | Neogene (Middle Miocene) | Gavdos Island Sheet | no |
| 11 | IACM | | - | - | - | Marls, marly limestones, clays | Neogene (Lower-middle Pliocene) | Heraklion Sheet | likely |
| 12 | IDI | | - | - | - | Ionian Zone (Aghios Yakinthos formation) | Upper Paleocene - middle Eocene | Anoyia Sheet | yes |
| 13 | IKRA | | Alluvial deposits | Mesozoic | - | Marbles, slates | Mesozoic? | * | yes |
| 14 | IMMV | | - | - | - | Marls, sandstones, conglomerates | Quaternary older | Vatolakkos (Alikianon) Sheet | no |
| 15 | ITM | limestone | Alluvial deposits | Quaternary - Holocene | claystone | Limestones with Rudistes | Upper Cretaceous (Santonien - Maestrichtien) | Meligalas Sheet | no |
| 16 | JAN | limestone | Alluvial deposits | Upper Jurassic - Lower Cretaceous | - | Old alluvian with fragments of silex from Vigla limestones and Doggerian sediments | Quaternary | Ioannina Sheet | no |
| 17 | KARP | | - | - | sandstone | Tripolitza series (Flysch: Marls, sandstones and conglomerates) | Tertiary / Upper Eocen (lower Priabonian) | South Karpathos Island Sheet | yes |
| 18 | KASA | Vigla limestone | Alluvial deposits | Upper Jurassic | limestone | Limestones of Vigla | upper Jurassic (Tithonion) - upper Cretaceous | North Korfou Sheet | yes |
| 19 | KEK | | - | - | limestone | Limestones of Vigla | upper Jurassic (Tithonion) - upper Cretaceous | North Korfou Sheet | yes |
| 20 | KLNA | limestone/alluvial deposits | Alluvial deposits | Holocene | - | Screes and fans with calcareous material | Quaternary (Pleistocene) | Kalymnos Sheet | no |
| 21 | KLV | | - | - | - | Pindos unit (platty limestones) | Upper Cretaceous | *Trikolas (2005) | yes |
| 22 | KSL | | Alluvial deposits | Paleogene | limestone | Paxos Zone (medium-thick bedded limestones) | Paleocene | Kastellorizo Sheet | yes |
| 23 | KSTE | | - | - | - | Autochthonus series of Crete-Ionian(?) Zone (Platty limestones) | Middle Jurassic-Eocene | Mochos Sheet | yes |
| 24 | KTHA | | - | - | - | Tripolis Zone (limestones) | Cretaceous (Undivined) | Kythira Sheet | yes |
| 25 | KVLA | limestone | Alluvial deposits | Oligocene | - | Kavala granite | Quaternary | Kavala Sheet | yes |
| 26 | KYMI | limestone | - | - | - | Subpelagonian - Pelagonian zone (limestones transgressive) | Cenomanian - Maestrichtian | Kymi Sheet | yes |
| 27 | KZN | limestone | Alluvial deposits | Cretaceous (Turonien-Maestrichtien) | sandstone | Pelagonian Zone (Formations of Kozani channel): calcareous material from transgression | Middle-Upper Cretaceous | Kozani Sheet | yes |



| 28 | LIA | peridotite | Alluvial deposits | - | - | Katalakon unit (lava domes, lava flows, breccias) | Neogene (lower Miocene - Aquitanian) | Limnos (Myrina) Sheet | yes |
|---|---|---|---|---|---|---|---|---|---|
| 29 | LKR | limestone | - | - | limestone | Postalpine sediments (graywackes, conglomerates, quartzites, shales, sandstones) | Paleozoic (Permian - Carbonniferous) | Livanatai-Atalanti Sheet | no |
| 30 | MGNA | limestone | limestones | Senonian | - | Ionian Zone (limestones with Rudist fragments) | Senonian | Kalamos Sheet | yes |
| 31 | MHLO | | - | - | - | Scree and fans | Quarternary | Milos Island Sheet | no |
| 32 | NEO | | - | - | - | Pelagonian Zone (Mica schists) | Preupper-Cretaceous tectonic nappe (Eohellenic tectonic nappe) | Zagora-Syki Sheet | yes |
| 33 | NISR | | - | - | - | Andesitic lavas and pyroclastics | Quaternary | Nisyros Sheet | yes |
| 34 | NISR2 | | - | - | - | Nikia rhyolite (Domes and lava flows) | Quaternary | Nisyros Sheet | yes |
| 35 | NOAC | | Limestone | Cretaceous (Cenomanian) | - | Allochthonus series (limestone hosting Fe-Ni pisolitic lateritic ores) | Cenomanian-Turonian | Athinai-Piraievs Sheet | yes |
| 36 | NPS | | - | - | - | Autochthonus series of Crete-Ionian? Zone (Platty limestones) | Middle Jurassic-Eocene | Ayios Nikolaos Sheet | yes |
| 37 | NVR | sandstone | Scree and Talus Cones | Pleistocene | - | Metamorphic rocks/upper series (schists, schist-gneisses, gneisses, amphibolites and marbles) | Oligocene - Miocene | Kato Nevrokopion Sheet | yes |
| 38 | ORTH | | - | - | - | Paxos Zone limestones with Rudist fragments and Foraminifera) | Upper Cretaceous (Santonian) | Zakinthos Island Sheet | yes |
| 39 | PENT | | - | - | sandstone | Sandstones and conglomerates of Pentalofos | Postalpine sediments (lower Miocene / Aquitanian) | Pentalofon Sheet | no |
| 40 | PLG | gneiss | Basal Conglomerate Series | Upper Miocene - Lower Pliocene | - | Quartzites and quartzitic sandstones of Svoula group | Triassic - middle Jurassic | Polygyros Sheet | yes |
| 41 | PRK | limestone | - | - | - | Extrusive Rocks (pyroclastic layer with lapilli tuff, tuff breccia or agglomerates) | Mainly Pliocene | Lesbos Island-Ayia Paraskevi Sheet | yes |
| 42 | PSRA | | Schists | Carboniferous-Paleozoic | - | Metapelites, Phyllites, meta-litharenites, metagreywackes | Mesozoic | *Meinhold et al. (2007) | yes |
| 43 | PTL | | - | - | - | Almyropotamos-Attiki autochthonous unit (marbles hosting Fe-mineral ore deposits) | Middle Eocene | Kifissia Sheet | yes |
| 44 | RDO | | - | - | - | Sediments and volcanics | Upper Eocene-Oligocene | Kardamos-Sapai Sheet | yes |
| 45 | RLS | | - | - | - | Gavrovo Zone (flysch with marls, sandstones and conglomerates) | Eocene | Nea Manolas Sheet | likely |
| 46 | SIVA | limestone | Conglomerates, sandstones, sands and marls or clays | Upper Miocene (Tortonian) | - | Series allochtones, internal zones, Asteroussia nappes (gneiss) | Upper Jurassic-Lower Cretaceous | Timbakion Sheet | yes |
| 47 | SKY | | - | - | - | Pelagonian Zone (limestone sequence) | Middle-Upper Cretaceous | Skyros Island Sheet | yes |





| | | | | | | | | |
|---|---|---|---|---|---|---|---|---|
| 48 | SMG | - | - | - | Metamorphic system (Vourliotes "Syrrachos" marbles) | Neogene | East Samos Island Sheet | yes |
| 49 | SMTH | Schist series | crumbled and symmetrically folded sedimentary rocks | Upper Jurassic/Lower Cretaceous | Ultramafic rock | Geological basement (slate series) | Upper Jurassic-Lower Cretaceous | Samothraki Sheet | yes |
| 50 | TETR | - | - | - | Gavrovo Zone (flysch undivided with sandstones and marls) | Priabonian-Oligocene | Mirofillon Sheet | yes |
| 51 | THERA | | - | - | - | Prevolcanic Sedimentary, Metamorphic and Igneous Rocks (crystalline limestones) | Upper Triassic | Thira Island Sheet | yes |
| 52 | THL | Limestone | - | - | - | Pelagonian Zone (marbles) | Middle Triassic-Lower Jurassic | Farkadon Sheet | yes |
| 53 | THVA | marles | - | - | - | Conglomerates, sandstones, sands, red loams | Pleistocene | Thivai Sheet | no |
| 54 | TNSA | | Greenschist | Permian | Felsic rock (Al) | Atticocycladic Complex, Upper unit (greenschists - prasinites) | Permian (?) | Tinos-Yaros Islands Sheet | yes |
| 55 | VAM | - | - | - | Marly limestone | Miocene | Chania Sheet | yes |
| 56 | VLI | - | - | limestone | Pelagonian Zone (carbonate rocks) | Upper Permian-Middle Triassic | Pappadhianika-Potamos Sheet | yes |
| 57 | VLMS | cretaceous limestone | Cretaceous limestone | Upper Cretaceous - Paleocene | | Paxos Zone: limestones with Rudist fragments and Foraminifera | Upper Cretaceous (Santonian) | Zakinthos Island Sheet | yes |
| 58 | VLS | alluvium and scree | Holocene | - | Alluvium and scree | Holocene | Cephalonia Island (Southern Island) Sheet | no |
| 59 | VLY | | - | - | limestone | Autochonus unit (dolomites Pirnaris) | Norian (upper Triassic) - Lias (lower Jurassic) ? | Koropi-Plaka Sheet | yes |
| 60 | ZKR | Limestone | Gray, dark-gray or black dolomite of bituminous odour when crushed | Triassic | - | Tripolitza series of Crete (limestones with Radiolites) | upper Cretaceous | Sitia (Ziros) Sheet | yes |

## 4 Discussion and conclusions

In the previous, we compiled several descriptors for our stations and derived amplification characteristics from our strong-motion data analysis. We now bring everything together to co-evaluate the overall potential of our stations as reference stations. We do not attribute numerical values and weights to each parameter, as is done e.g. in the summation rationale of

350 Lanzano et al. (2021). We believe there are inherent issues with quantifying qualitative data and treating them as homogeneous to perform mathematical operations between them. Moreover, our goal is not to provide a continuous ranking across all sites. We opt for co-assessing all input and offering an overall qualitative assessment of reference site potential. In Table 7 we consider stations that got a positive assessment in all 5 factors as 'preferred' reference sites (2 instances), those who missed 1 field as 'very good' (3 instances), and those that missed 2 fields as 'ok' (18 instances). Stations that ranked



lower are not recommended, though the user can select them for specific purposes or within specific frequency bands according to her/his own judgement. Different schemes could be contrived to evaluate and even prioritise the stations, but we do not feel an absolute grading is necessary, especially since the appropriateness will also depend on the precise nature of the application making use of the reference motion. It is a strong message for us to convey that over half the stations did not rank as reliable enough reference stations, and we feel that more work is needed to reassess the implications of this finding. It is

also interesting to note that several of our rock sites had high-frequency amplifications: this is in line with the definition of A-class sites in EC8, which is shifting from the current version (CEN, 2004) of $Vs_{30}$>800 m/s, to a new version (Labbé and Paolucci, 2022) where there is also a provision of $f_0$>10 Hz.

In this study, we compute FAS-based HVSR for the first time for all the HL rock stations, producing $f_0$ and other metadata. We also compile all existing parameters we can find from various sources (housing/installation, topography/slope, surface

geology, and $Vs_{30}$; ad-hoc Vs profiles being almost non-existent across Greek seismic rock stations). We compare and contrast those metadata from various sources and, in addition, we offer insights and corrections based on site visits from a network operator's point of view. We believe this operator's first-hand experience is important because geological maps constructed at such a scale as to serve an entire country (and made by different teams, over several decades) inevitably contain errors and simplifications, whereas a site walkover of the station location by an experienced geologist provides

additional reliability. Similarly, satellite-based estimates of slope/topography invariably include approximation, homogenisation and some lack of specificity depending on the size of the 'pixel', whereas again a site visit leaves little doubt as to the exact nature of the landscape at the exact location of the station. The information for rock stations up to now has been sparse and scattered for the strong-motion case, and almost nonexistent for the broadband one. Until now, if a user wished to select a reference station in the HL network, s/he might have resorted to geology, or even considered all rock

stations as interchangeable. We hope this work has provided the first step towards a better evaluation of rock stations and eventually towards the better utilisation of their data.

Finally, we believe that data-derived transfer functions are extremely important and illuminating for understanding station response. There is sometimes a fixation on $Vs_{30}$ which is not only inadequate (too shallow, and providing no indication of impedance depth or contrast), but may even be unnecessary if we have both the geology and –what is more- the empirical

site response from recordings. Even a full Vs profile may be inadequate to fully assess site response, if we consider that its high-frequency part depends heavily on the assumptions we made of damping, and –most of all- that its premise for yielding reliable site response is that the 1D assumption holds true, which in nature is rarely the case (and especially perhaps for rock sites - whereas empirical estimates of site effects, may have their shortcomings but reflect the 3D nature of the formations). Our study has shown once again that not all 'rock' sites should be treated -or trusted- equally. Also, we would ask the

question: if we have data-derived site response, how much importance should stand-alone meta-descriptors and proxies such as $Vs_{30}$ be given?





**Table 7**. Compilation of reference site potential per station according to all factors and final disposition.

| No. | Station code | Topography | $Vs_{30}$ | Geology | HV shape & level | Directionality | Final disposition |
|---|---|---|---|---|---|---|---|
| 1 | AMGA | yes | no | yes | yes | yes | ok |
| 2 | ANKY | yes | yes - ESM | likely | no | | |
| 3 | APE | yes | - | yes | no | | |
| 4 | ARG | yes | no | no | ok | | |
| 5 | ASTA | yes | yes - ESM | no | yes | | ok |
| 6 | ATHP | yes | yes - Helpos | no | yes | | ok |
| 7 | DION | yes | no | yes | ok | | |
| 8 | DLFA | no | yes - ESM | yes | yes | yes | very good |
| 9 | EVR | yes | - | yes | yes | | ok |
| 10 | GVD | yes | no | no | no | yes | |
| 11 | IACM | yes | no | likely | no | | |
| 12 | IDI | no | - | yes | yes | yes | ok |
| 13 | IKRA | yes | no | yes | ok | | |
| 14 | IMMV | yes | yes - ESM | no | yes | | ok |
| 15 | ITM | yes | yes - ESM | no | ok | | |
| 16 | JAN | yes | no | no | ok | | |
| 17 | KARP | no | yes - ESM | yes | yes | yes | very good |
| 18 | KASA | no | yes - ESM | yes | yes | | ok |
| 19 | KEK | no | - | yes | ok | | |
| 20 | KLNA | yes | no | no | no | | |
| 21 | KLV | no | - | yes | yes | yes | ok |
| 22 | KSL | no | yes - ESM | yes | ok | | |
| 23 | KSTE | no | - | yes | ok | | |
| 24 | KTHA | no | no | yes | no | | |
| 25 | KVLA | yes | yes - ESM | yes | yes | yes | preferred |
| 26 | KYMI | no | yes - ESM | yes | ok | | |
| 27 | KZN | yes | no | yes | ok | | |
| 28 | LIA | yes | no | yes | ok | | |
| 29 | LKR | yes | - | no | no | | |
| 30 | MGNA | yes | no | yes | yes | | ok |
| 31 | MHLO | yes | - | no | no | | |
| 32 | NEO | yes | - | yes | no | | |
| 33 | NISR | no | - | yes | no | | |
| 34 | NISR2 | no | - | yes | no | | |
| 35 | NOAC | yes | no | yes | ok | | |
| 36 | NPS | yes | - | yes | yes | | ok |
| 37 | NVR | yes | yes - ESM | yes | ok | yes | preferred |
| 38 | ORTH | yes | - | yes | no | | |
| 39 | PENT | no | - | no | yes | | |
| 40 | PLG | yes | no | yes | yes | yes | ok |
| 41 | PRK | yes | no | yes | no | | |
| 42 | PSRA | yes | no | yes | no | yes | ok |
| 43 | PTL | yes | - | yes | yes | | ok |
| 44 | RDO | yes | - | yes | ok | | |
| 45 | RLS | yes | - | likely | ok | | |
| 46 | SIVA | yes | no | yes | no | | |
| 47 | SKY | no | - | yes | no | | |
| 48 | SMG | yes | no | yes | no | | |
| 49 | SMTH | no | yes - ESM | yes | yes | | ok |
| 50 | TETR | no | - | yes | no | | |
| 51 | THERA | no | - | yes | yes | | |
| 52 | THL | yes | - | yes | yes | yes | very good |
| 53 | THVA | yes | - | no | no | | |
| 54 | TNSA | yes | no | yes | yes | | ok |
| 55 | VAM | yes | - | yes | ok | | |
| 56 | VLI | yes | - | yes | no | yes | ok |
| 57 | VLMS | yes | no | yes | no | | |
| 58 | VLS | yes | yes - Marg21 | no | yes | | ok |
| 59 | VLY | yes | - | yes | ok | | |




| 60 | ZKR | yes | yes - Marg21 | yes | no | ok |

**Data availability**

All waveforms and station metadata were downloaded and are freely accessible at https://eida.gein.noa.gr/, the regional node of EIDA (the European Integrated Data Archive) hosted by the Institute of Geodynamics of the National Observatory of Athens (NOA). Data from NOA's seismic network bear the network code HL and are attributed DOI:10.7914/SN/HL. Event parameters come from the seismic catalogue of NOA, freely accessible here: https://eida.gein.noa.gr/fdsnws/availability/1. Station metadata come from the various articles cited in the paper, as well as the ESM (https://esm-db.eu/; Lanzano et al.,
2021; Luzi et al., 2016). Maps published by the Hellenic Survey of Geology and mineral Exploration are generally available by HSGME for purchase and hence not freely accessible.

**Author contribution**

OJK had the idea, coordinated the team, and wrote the paper. AP performed manual data processing and final data check and curated the database. EVP performed the main coding and calculations. ZC compiled and interpreted the ground motion
station metadata. FG performed manual data processing. SL contributed the geological interpretations and oversaw PS and KF in compiling the geological station metadata. CPE provided insights on station history, installation and performance.

**Competing interest**

The contact author has declared that none of the authors has any competing interests.

**Acknowledgements**

Discussions with several colleagues have helped the lead author since she joined NOA in 2018; among others, Ioannis Kalogeras, Thymios Sokos, Hiroshi Kawase, Laurentiu Danciu and Alexis Chatzipetros are cordially thanked in order of appearance. No funding is acknowledged, except the small internal scholarship 'ROAR' awarded by NOA to OJK. Some plots were created using Generic Mapping Tools (GMT; Wessel et al. 2013) and the help of this fantastic tool is cordially acknowledged. Finally, no AI was used in developing this manuscript; we used what human intelligence was available.





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
