# Peer review of "The quest for reference stations at the National Observatory of Athens, Greece"

_Natural Hazards and Earth System Sciences, 2023_

## Referee Comment (RC1)

This study presents a compilation of mostly existing site information in search of reference sties in Greece. It collects some important site data which are crucial for may downstream studies, serving as an important dataset for Greece and the international community. The manuscript is relatively well written, and I enjoyed reading it very much. Most of the information here is collected from existing sources, in my view, addition efforts could be devoted to uniformly deriving some extra information, e.g., empirical site response, site kappa, and topographic parameters, to render the dataset even more valuable. The following are my specific comments:

**Major comments:**

1. GIT or deltS2S site response: since waveforms are available, why don't the authors derive site response from observations using either generalized inversion or residual analysis to the median prediction from a ground motion model (GMM)? These two techniques are equivalent and can give same results when the same constraints are applied. Though reference sites need to be assumed, it can be used as the average site response over all sites in the dataset. Consequently, the resulting site responses are relative to this condition. However, in my view, these observed (relative) site responses carry the most important information in terms of reference site selection. A good reference site shall exhibit smooth change in amplitude with frequency, thus teasing out sites with significant resonance (with clear peaks or throughs).
Though HVSR is used here, it is well known, and is clearly pointed out by the authors, that HVSR only approximates (horizontal) site response at frequency range where there is no significant vertical site response. However, we know as little in which frequency range that vertical site response is negligible at a specific site, as the horizontal site response.
Thus, it is valid that: a good reference -> flat HVSR over broad frequency range, however, invalid: flat HVSR → a good reference. I just provide one real example (KiK-net site) below. The HVSR seems to be relatively flat, which is because the horizontal peak is canceled by the vertical.
Therefore, I suggest the authors to consider using GIT or residual analysis to derive empirical (relative) site response to compliment HVSR, as did Lanzano et al. (2020).
For this very reason, consider to use "Earthquake HVSR" directly, rather than 'Transfer Function' as section title. The former is clearer to readers.

[Figure]

2. Site kappa: likewise, given available waveforms, I think site kappa can be another source of information for site selection. Please consider deriving site kappa at these sites.
3. Data availability: I strongly suggest providing the information in various tables in the paper in a single flatfile (e.g., csv) as electronic supplemental materials, which will greatly facilitate the use of these results by end users. Ideally, only symbols in the table header that are readable to machines can be used such that the table can be read directly by computers.
4. On HVSR computation: it is mentioned that the two horizontal components are combined as the square root of the sum of squares, rather than the root-mean-squares. Thus, would HVSR be unity, or sqrt(2), even at perfect rock sites? This is important for the following statements in Line 150.
5. Table 4: some sties lack topographic data, i.e., slope. There topographic data can be readily derived from openly available DEM. I suggest the authors devote some efforts to deriving them.

**Minor comments:**
1. Line 225: S1 should be A1?

I have happy to waiver my anonymity.

Best regards,
Chuanbin

---

## Author Comment (AC1)

1. Although this study makes a significant contribution to the understanding of reference station conditions for rock sites in the broadband and accelerometric networks of the National Observatory of Athens, it is important to acknowledge a key limitation.

➜ We are glad that our work instigated interest by members of the broader community, and would like to take the opportunity to clarify some misconceptions in what follows. Most of the points labelled as shortcomings or limitations in this commentary are according to our rationale some of the notions that lend our paper its strength, necessity and timeliness.

2a. The study relies heavily on existing data from publicly available sources and past studies

➜ No previous study has ever brought together ad hoc all existing information for the stations of the HL network, the only network spanning the entire national territory of Greece and comprising both broadband and strong-motion stations. Compiling the information is the first necessary step towards effectively assessing it and improving its current understanding and use. The importance of comparing and scrutinizing publicly available sources is well in line with current FAIR principles governing data sharing in Europe, and endorsed and promoted by European entities such as EFEHR. This process promotes assessment of the data quality and value, ultimately benefiting transparency and usability/reusability, helping improve the quality of future studies.

2b. and the selection of stations is based on the belief that they are situated on rocks.

➜ The very essence of this work is to closely scrutinise and challenge any established beliefs as to the adequacy of the reference conditions that may have been hitherto presumed based on simplified criteria such as broad geological descriptions. The stations likely to be used as reference are precisely those for which the belief exists that they lie on rock, and those are the ones we feel it is most urgent to examine. We could easily have included accelerometric stations that are thought to lie on alluvia, but there is absolutely no reason to fear that any cognizant scientist would choose them as reference stations. Finally, we should clarify that the notion of 'belief' is not based on one source, but is inclusive: we scan all available sources (ESM, accelnet website, publications) and if even one of them makes mention to rock then we select and study said station.

3. This report exhibits several notable shortcomings that may impact the validity and reliability of the findings. First, the reliance on an established belief that the selected stations are situated on the rock without conducting prior site effect studies raises concerns about the accuracy of the assumed geological conditions.

➜ Precisely because there exist no prior site effects studies for the HL network as a whole, this is the first time a site effects study is performed in this paper. We do this by means of constructing a curated, high-quality strong-motion dataset and analyzing amplification with the non-reference method of HVSR.

We have no concerns as to the accuracy of the assumed geological conditions: on the contrary, we openly challenge any assumptions by confronting them with what the recorded data show, allowing for the first time the data -dozens or hundreds of recordings per station- to do the talking. See also reply to point 2b.

4. Additionally, the absence of ad hoc field campaigns for characterizing the stations, except in two cases, introduces a significant limitation in understanding geological units and age, as well as other critical characteristics.

➜ It is precisely because we do not have access to the very large resources one would need to characterize each and every station in situ –especially considering the geographical distribution and challenging geomorphology of Greece, including many distant islands and remote regions-, that we see evident value in making use of the freely available seismic data recorded over the past two decades. While the exhaustive field campaigns proposed in CC1 would essentially require a new and large nationwide investment in terms of time, personnel and funding, our approach has actually just made use of an old, yet still large, investment that was already made in our country, namely the instrumentation and network operation, and which had essentially been waiting in the proverbial drawer all this time. In our mind, it would be unimaginable *not* to take the opportunity to do this work. Moreover, we would like to clarify that seismic-data-derived analyses are not just the poor man's (or woman's!) alternative to the proposed in-situ characterisation methods: on the contrary, all methods have their limitations. Site campaigns are very welcome when the resources exist for them, but empirical spectral ratios -in the various ways in which they are used here- can sometimes even yield further information, simply because they are unhindered by certain assumptions inherent in campaigns. For instance, here we allow the data itself to indicate any directional dependence of the site response by analysing amplification for rotational increments: this allows for potential 2D/3D phenomena to become manifest. On the contrary, standard geophysical campaigns would yield a preferred 1D profile (the most probable one among a suite of possible solutions), based on which

forward modeling would compute a 1D transfer function that would in turn afford no indication of directionality.

Finally, we should clarify that, in our view, in-situ campaigns could not possibly allow for any further understanding in the geological units or age per se (we are unsure what other 'critical characteristics' our commenter has in mind), since they would focus on wave propagation-related properties, namely velocities, and in an ideal world perhaps damping. What could indeed help improve our understanding of geological units and age could be the drilling of boreholes at all network stations, as is the case e.g. for the kik-net; but this would constitute an even larger nationwide investment, one that we can certainly wish for but are not prepared to wait for.

5. The use of ad-hoc information from maps and operator experience while attempting to enhance site descriptions may introduce subjective biases and lack the rigor of systematic field studies.

➔ On operator experience: Information coming from the only specialised personnel to ever visit the station on behalf of the operator may be 'subjective' in that it comes from a human being, but we consider it as lying much closer to expert opinion than to bias. We'd also like to point out that the idea of retrieving and preserving first-hand field information gathered from the operator's side was also proposed by us within the Engineering Seismology CRG of the ongoing AdriaArray project and was endorsed as a practice to be encouraged and systematically applied throughout its hundreds of stations.

On maps: Information coming from maps may conceivably carry the subjectivity of the respective specialists who were employed by the competent national authority to make the map (a risk we are prepared to take, given that any data ever compiled may be blamed in the same way following such a trail of thought), and the bias of scale and of decades gone by since the mapping (a bias that is more likely, in our opinion): but this is precisely the reason why we carefully seek out that information, document it, and then confront it with all other data types we could find. This practice actually minimizes the impact of any single bias by considering all possible data, accounting essentially for epistemic uncertainty. A systematic field study, even if it were possible, would after all carry its own biases and uncertainties.

6. Furthermore, the report acknowledges the absence of previous site effect studies for the ensemble of stations under investigation, suggesting a potential gap in foundational understanding.

➔ We suggest there exists a definite gap in site effects knowledge, and we proceed to partially rectify that. We do not know what is meant by 'foundational understanding' or gap thereof.

7. The reliance on publicly available data and compilation of existing information may lead to incomplete or outdated datasets, compromising the overall robustness of the analysis.

➔ We believe that scrutinising and compiling data can only help towards more complete and updated information and cannot pose any kind of threat to any analysis.

8. The report's recommendation of preferred reference sites is contingent on the assumptions and methodologies employed, raising questions regarding the generalizability and applicability of the findings to broader hazard applications.

➔ Our recommendations, as is the case with any conclusion of any work, are indeed contingent upon our assumptions and methods – were they not so, they could be criticised as being arbitrary or unjustified. The method we proposed is in itself simple and easily generalisable to any region or network, since no assumption has been made that limits applicability to Greece or the HL. The data processing itself has been performed precisely in the spirit of engineering seismology and hazard applications, following upon the footsteps of the PEER framework, which the main author helped shape.

9. Overall, these limitations underscore the need for a more comprehensive and rigorous approach to ensure the credibility of a report's conclusions. While the article compensates by combining available information, including operator experience and ad-hoc data, it highlights a potential gap in the comprehensive understanding of the geological and site-specific features of these stations. Future research could benefit from targeted field campaigns to fill this gap, enhance the robustness of the findings, and provide a more accurate assessment of the suitability of the stations as reference sites.

➔ We only partially understand this paragraph. We can only wish that in future there may be large, targeted and systematic investments in in-situ characterisation, including both intrusive and nonintrusive techniques. In the meantime, between awaiting potential new investments and doing something now to capitalise on existing ones, we opt for the latter.

10. It should not be accepted as a research article than a report in such a highly acknowledged journal such as Natural Hazards and Earth System Sciences.

---

## Author Comment (AC2)

This study presents a compilation of mostly existing site information in search of reference sties in Greece. It collects some important site data which are crucial for may downstream studies, serving as an important dataset for Greece and the international community. The manuscript is relatively well written, and I enjoyed reading it very much. Most of the information here is collected from existing sources, in my view, addition efforts could be devoted to uniformly deriving some extra information, e.g., empirical site response, site kappa, and topographic parameters, to render the dataset even more valuable. The following are my specific comments:

➔ We thank the reviewer for their positive feedback. We appreciate his suggestions and agree with many of them, but will endeavor to explain in what follows why most of the additional studies proposed are at this moment out of the scope of the present work.

Major comments:

1.
GIT or deltS2S site response: since waveforms are available, why don't the authors derive site response from observations using either generalized inversion or residual analysis to the median prediction from a ground motion model (GMM)? These two techniques are equivalent and can give same results when the same constraints are applied. Though reference sites need to be assumed, it can be used as the average site response over all sites in the dataset. Consequently, the resulting site responses are relative to this condition. However, in my view, these observed (relative) site responses carry the most important information in terms of reference site selection. A good reference site shall exhibit smooth change in amplitude with frequency, thus teasing out sites with significant resonance (with clear peaks or throughs).

➔ We really like the techniques mentioned by the reviewer and will be very happy to test them at a later date, after our database has been curated with respect to all the necessary parameters. At this time, it is not straightforward to implement neither a broadband spectral inversion for source, path and site components, nor a full residuals analysis with respect to GMMs. Both these techniques would necessitate further curation and homogenization of non-site parameters, most notably catalogue parameters such as magnitudes and fault mechanisms. Both GIT and ds2s can suffer from trade-offs and need the user to exercise care so that e.g. source-related factors do not map onto site-related (or even path-related) ones and vice versa (e.g. stress-drop vs moment or path attenuation vs site amplification in the one case, τ vs φ trade-offs in the other case, etc.). For a successful result, we believe it is also generally desirable to have a well-balanced source-station distribution with as many events as possible being simultaneously recorded at as many stations. None of these issues or limitations really hinder HVSR; this is why we believe that spectral ratios are indeed the first technique we should be using in this new dataset we compiled, before delving into less straightforward ones that require additional checks, experience and judgement for correct interpretation. Thus, we would like to respectfully save the reviewer's recommendations for when we are ready to move on to a full inversion and/or residuals analysis, which will constitute a new research step. We also point out that adding to the scope of the work performed implies additional resources.

Based on the reviewer's recommendations for these additional analyses, we are concerned that perhaps our work may have struck the reviewer as somewhat limited in scope or simple. In an effort to rectify this impression, we would like to permit ourselves to point out a few considerations:

1. A large part of this work comprises the compilation/curation of the database itself. It might perhaps be easy to pass by, yet it actually took over a year to carefully analyse the waveforms with the in-house software we developed ad hoc for this purpose. It was an iterative (and rather painful, despite having some significant past experiences in strong-motion processing) procedure, during which our processing procedure was improved, the data reanalysed, and several errors in the data or metadata were discovered, investigated and eventually amended. We do not claim to have done a perfect job, but our experience taught us that creating a database from scratch to perform analyses is truly challenging compared to using existing curated ones. We would thus like to respectfully draw some attention to the value we feel this effort carries in its own right.

2. Although our empirical analysis only includes HVSR, we would like to point out that we did try to use the technique in as exhaustive a way as possible, including features that are –in our experience- quite rarely explored, such as:

a. going to great pains to control the usable bandwidth within which it is computed, so as not to attribute significance to results where the data are too noisy;

b. carefully examining the full range of rotational possibilities, so as to sample the site effects not only at the as-recorded directions but in all possible directions (since geomorphological features do not have to oblige by following the orientation of the sensor), so as to find the maximum amplification in

whichever direction it takes place, and even assessing the rotation-related variability of the amplification over selected frequency ranges in a quantitative way, which is something not done before -to our knowledge.
c. considering the known fact that HVSR underestimates the amplification level and likely the higher modes, we implemented the correction method of Ito to provide an approximate estimate of the potential amplification level that could be expected.

Though HVSR is used here, it is well known, and is clearly pointed out by the authors, that HVSR only approximates (horizontal) site response at frequency range where there is no significant vertical site response. However, we know as little in which frequency range that vertical site response is negligible at a specific site, as the horizontal site response.
Thus, it is valid that: a good reference -> flat HVSR over broad frequency range, however, invalid: flat HVSR → a good reference. I just provide one real example (KiK-net site) below. The HVSR seems to be relatively flat, which is because the horizontal peak is canceled by the vertical.

[Figure]

Therefore, I suggest the authors to consider using GIT or residual analysis to derive empirical (relative) site response to compliment HVSR, as did Lanzano et al. (2020).
For this very reason, consider to use "Earthquake HVSR" directly, rather than 'Transfer Function' as section title. The former is clearer to readers.
➔ We thank the reviewer for this comment. We agree that a flat HVSR does not necessarily a perfect reference site make, though the reverse should be true. However, we'd allow ourselves to comment that the plot given as an example of HVSR limitations seems a bit extreme to us: it shows a site whose amplification according to SSR seems to reach 10^1.5=32 at 10 Hz. We have never come across such a site, and although we do not doubt the correctness of this plot, we do believe it describes a somewhat special case. (We also wonder whether the horizontal downhole FAS may not have a sharp trough at 10 Hz that might perhaps cause an exaggerated peak of surface SSR -- in other words, is the surface being divided with a 'perfect' reference site or could this instance be affected by the limitations of the SSR method itself? But this is outside the scope of the discussion, only a passing thought.)
Though we have no means of deriving SSR for Greece, we are happy to include a stronger 'disclaimer' in our manuscript as warning that even 'Ito-corrected' HVSR amplifications may still underestimate resonance, but we honestly do not expect such a shocking case among our stations. We'd also like to mention that, although we neglected to include it in our list of references, we are acquainted with the reviewer's recent work of Zhu et al. (2020) entitled 'Detecting Site Resonant Frequency Using HVSR: Fourier versus Response Spectrum and the First versus the Highest Peak Frequency' and we agree -both in theory and in practice, i.e. in this paper- with many of its conclusions. Their conclusions included propositions such as: that FAS-based HVSR yields an estimate of where the maximum site amplification takes place, even if it provides a lower bound; that network operators should make public their fp frequency estimates for their stations as a matter of priority, and preferably their entire HVSR curves for users to scrutinise and reach their own assessment of f0/fp. This is what we have tried to do. We agree it is intriguing to further investigate vertical amplification, but at this stage in our work, we feel we should leave that for the future.
We can certainly use the term 'Earthquake HVSR' rather than 'Transfer Function' in the paper, if the latter encourages a misapprehension that it is free from the HVSR limitations.

2.
Site kappa: likewise, given available waveforms, I think site kappa can be another source of information for site selection. Please consider deriving site kappa at these sites.
➔ Thank you for this comment, which was also raised by Reviewer 2. We would certainly like to study kappa for the stations, but will need to construct a different database for this purpose. Plots S1 and S2 show the magnitude-distance distribution of our sets per station, and because of our criterion for M4 and above (and the fact that HH stations can clip in the near field), most stations do not have nearby

recordings. For Greece, 'nearby' can mean within 20-25 km, as in Ktenidou et al. (2015 - https://doi.org/10.1093/gji/ggv315) we showed that there can be quite an increase of $\kappa_r$ with distance after 30 km. Only a handful of our stations have more than a couple of data points within 30 km. For the vast majority of them an extrapolation of $\kappa_r$ to R=0 would yield a significant uncertainty in $\kappa_0$, as the latter would be rendered strongly dependent on the path attenuation being removed. So again we stumble upon the issue of site-path trade-offs, and to resolve this to a satisfactory extent we feel we must address it in a future step with an improved dataset.

3.
Data availability: I strongly suggest providing the information in various tables in the paper in a single flatfile (e.g., csv) as electronic supplemental materials, which will greatly facilitate the use of these results by end users. Ideally, only symbols in the table header that are readable to machines can be used such that the table can be read directly by computers.
➔ We can certainly try to do that to facilitate the uptake of our results by readers (and machines!).

4.
On HVSR computation: it is mentioned that the two horizontal components are combined as the square root of the sum of squares, rather than the root-mean-squares. Thus, would HVSR be unity, or sqrt(2), even at perfect rock sites? This is important for the following statements in Line 150.
➔ The reviewer has a point. We can amend the manuscript to clarify the effect of how components are combined.

5.
Table 4: some sties lack topographic data, i.e., slope. There topographic data can be readily derived from openly available DEM. I suggest the authors devote some efforts to deriving them.
➔ We can look into that.

Minor comments:
1.
Line 225: S1 should be A1?
➔ Yes indeed, thank you for catching that.

I have happy to waiver my anonymity.
Best regards,
Chuanbin

---

## Author Comment (AC3)

The manuscript studies for the first time the conditions of reference stations across 60 rock stations belonging to the broadband and accelerometric networks of the National Observatory of Athens. The analysis is based on selecting stations situated on rock and whose data availability is sufficient for a meaningful collection of recordings. The study is relevant since no systematic previous site effects studies were conducted for the ensemble of stations under examination. Similar studies have been conducted in other contexts and at different scales by Lanzano et al. (2020, 2022) for Italy and by Pilz et al. (2020) for Europe. I believe that this work is a very useful contribution to the scientific community and to seismological studies in Greece. I suggest accepting the publication with minor revisions.
➔ We thank the reviewer for the overall appreciation of our rationale and outcomes, and suggested improvements.

Some comments below:

**Station and data selection**
1.  Please explain further, on the basis of which criterion you consider the 60 stations to be installed on rock. Did you use geological and/or topographical proxies? Do any of the stations also have a geophysical survey?
    ➔ One of the main problems is that few strong-motion stations in HL have had the benefit of geophysical studies, and of those that have, most are either old (triggering mode) or lie on soil deposits, so they were of no interest to us at this point (the scope being modern continuous data on potential rock). No broadband stations have been characterized, as is the global standard practice.
    In selecting the stations, these were the criteria: 1. HH and collocated HH/HN stations: we selected all of them; 2. HN stations: we selected a HN station if it was thought to lie on rock according to any one of the following 3 criteria: by the network operator, as per existing information on accelnet.gein.noa.gr website; by ESM, based on proxies as rock; and according to the detailed investigation of surface geological conditions performed for all HN stations using the HSGME maps.
    The reviewer is correct in that we did not explain the selection criteria clearly enough and this can be amended in the revision.
2.  Letters a and b are not present in Figure 1 and 2
    ➔ Yes indeed, thank you for catching that.

**Creation of a new strong motion dataset**
The analysis of the signals for the creation of the dataset is a very important step. I realised that the authors did a very thorough job, taking advantage of already available codes. However, I suggest that this section be reorganised schematically by indicating the data processing work in steps.
For example:
- Identification of clipped records (which criteria?)
- Waveform picking
- Calculation of signal-to-noise ratio
- Identification of corner frequencies - Etc.
I am not saying to repeat things that have been explained in other works, but to list the actions in a schematic and sequential manner. I think the work would benefit in terms of clarity. I think it would also be helpful to understand which signal analyses are done by the NGA-East code and which are not.
➔ We thank the reviewer for acknowledging the work load implied in creating this dataset. We also realize that we were not clear enough in our manuscript and created a misconception: we did not use any available off-the-shelf codes from PEER or elsewhere. We may have caused some of the confusion by mentioned PEER processing standards, but what we meant was that we were affected to a great extent by the logic followed in PEER NGA-West2 and East projects, where the first author was part of the data processing development team. But in this work, we created our own in-house processing code from scratch to analyse our data in the time and frequency domain for our specific purpose. We actually developed our code further as we processed more data, understood the needs arising and implemented improvements. We realize that the suggestions made by the reviewer will clarify the procedure we used and we can adopt them to better explain the flow and tools used.
Do you check for double events recordings? How do you treat?
➔ We are not sure we understand what the reviewer means by double recording: a recording of an event in whose tail another event takes place? In such cases, we try to salvage the recording if possible on a case-specific basis. If we understand the question better, we can address it in the manuscript.

I also suggest improving figure 3 to make it more self-explaining. A legend is missing.

➔ We can certainly do that.

**Transfer function**

1. For the calculation of transfer functions with the horizontal-to-vertical spectral ratio method, I would pay attention to the recordings of co-located stations.

From what I understand, all recordings for stations equipped with accelerometer and velocimeter were considered for the estimation of HVSR. My experience with INGV's Italian seismic network, which has a large number of collocated stations, is that this should be done carefully. I suggest conducting a preliminary analysis by keeping the instruments separate (consider them as two different stations) to verify that the transfer function is equal. First of all, the sensors could suffer from fixed scaling (a comparison of intensity measurements of recordings of the same event is also recommended) caused by incorrect conversion constants in the station xml. In addition, Hollander et al. (2020) and Castellaro et al. showed that the behaviour could be different at specific frequency ranges (especially in high frequency) due to different station installations.

➔ The reviewer makes an important comment here. We did not mix data from different channels (HH with HN) for any station analysed. When both were available, we selected the strong-motion one. Although a systematic comparison of the two sensor transfer functions is out of the scope of this particular article, we did make some cursory comparisons and these actually helped us identify a couple of mistaken sensor responses that were eventually corrected. In our case, if the gain is wrong on all three components in the same way, HVSR is unable to detect that, but the rotational sensitivity tests can indeed help identify cases where the N and E component are not consistent. Though an exhaustive account of all comparisons cannot fit in this paper, we can certainly address this question better in the manuscript, and we will take these suggestions on board for more detailed future work. The same holds regarding the useful comment of comparing high-frequency content from the two sensors in the light of installation differences.

1. I think it is also useful to explain how the groupings in Figure 7 were made. Was a clustering analysis conducted? Or is it based on a visual analysis of the curves? I have the impression that the transition from one group to another may not be clearly defined and some stations may be in one group rather than another arbitrarily. Wouldn't it be useful to set a quantitative criterion to isolate stations that have a flat response?

➔ The results shown in the submitted paper were grouped by visual inspection. This is not an optimal way to go about this. In the revision we will test some automated ways to yield the groups of stations, such as hierarchical clustering.

**Discussion and conclusions**

I noticed that the authors never considered the high-frequency near-site attenuation parameter k0 among the parameters for identifying reference sites. Considering the great experience of the authors, do you think this parameter could have any weight in the future? For example, in the work of Morasca et al. (2023) we started from the 36 stations of the study by Lanzano et al. (2020) and restricted to 6, as reference sites for a GIT in central Italy. The selection was made on the basis of k0, identifying those with k0<0.015s.

➔ The reviewer is correct - thank you for sharing your experiences with us. We appreciate the suggestion and vote of confidence, and we certainly wish to look into κ in future. However, as we responded to Reviewer 1 who had the same question -see point 2 above-, we need to reconsider the dataset and focus on an appropriate distribution that will include more near-source data to avoid results being overly dependent on path and distant events. When this is available, we can certainly try to use κ0 values as an additional piece of information when assessing reference stations.

**Additional reference**

Castellaro, S., Alessandrini, G., & Musinu, G. (2022). Seismic station installations and their impact on the recorded signals and derived quantities. Seismological Society of America, 93(6), 3348-3362.

Hollender, F., Roumelioti, Z., Maufroy, E., Traversa, P., & Mariscal, A. (2020). Can we trust high‐frequency content in strong‐motion database signals? Impact of housing, coupling, and installation depth of seismic sensors. Seismological Research Letters, 91(4), 2192-2205.

Morasca, P., D'Amico, M., Sgobba, S., Lanzano, G., Colavitti, L., Pacor, F., & Spallarossa, D. (2023). Empirical correlations between an FAS non-ergodic ground motion model and a GIT derived model for Central Italy. Geophysical Journal International, 233(1), 51-68.

➔ Thank you for sending these.

---

## Author Response (AR1)

Dr Olga-Joan Ktenidou
Senior researcher
Institute of Geodynamics
National Observatory of Athens
PO Box 20048, 11810, Athens, Greece
olga.ktenidou@noa.gr

Dr L. Danciu:
Guest Editor, Special issue on European Seismic Hazard and Risk models

Athens,
5/10/2023

Dear Editor,

Please find enclosed the revision of our work entitled 'The quest for reference stations at the National Observatory of Athens, Greece'.

We would like to sincerely thank the reviewers who gave us constructive comments and suggestions, as well as yourself. In what follows, we describe the main changes made to the manuscript. These are also clearly marked on the annotated documents using 'track changes'.

Thanks again for your time and consideration of our work. We are looking forward to hearing from you.

Yours sincerely, for the authors,

Olga-Joan Ktenidou
* * *
AUTHOR RESPONSE TO ACCOMPANY REVISIONS:

Dear Reviewers,
We would like to sincerely thank you for the constructive comments and suggestions, which helped improve this article considerably, as they led to an extensive revision in terms of content as well as presentation.
Below we summarise the changes we made to the manuscript as a result of your comments. We have addressed all comments and implemented almost all of them (where we did not, detailed explanations and rationale are given in our point-to-point response, this description of changes, and the amended manuscript.
With many thanks,
Olga Ktenidou, for the authors

Major changes:

- We added 2 years of data to all stations: 1/1/2022-31/12/2023, to make the results as up-to-date as possible. This increased the total Nrec from 6840 to 7512 and the mean Nrec per station up to 125. All analyses/interpretations/visual inspections etc. were redone for the new populations, and as a result all tables and figures were updated. An additional author is added due to this extra work.
- We rewrote the manual data processing steps extensively and improved fig. 3 to explain it. We clarified all novel/in-house aspects and differences from existing workflows, to highlight this

paper's contribution to making a new database with its own consistent and targeted processing flow.

- We rewrote the entire empirical HVSR part, creating subsections for the study of the mean HVSR, the rotations and sensitivity study, the VACF correction, and the clustering.
- We redid the clustering in a proper and systematic way, trying various algorithms, and in the end choosing hierarchical agglomerative clustering. We decided to cluster not only for the shape/level of HVSR, but also the VACF-corrected one.
- Figures 5 and 6 are added, which show a mosaic of some of the most and least 'passable' HVSR results based on mean shape/level. We decided that such a visual illustration would benefit the reader and enhance readability.
- The issue of sqrt(2) (1.4 as opposed to unity) that was brought up is discussed in detail, and the criteria for selecting the f0 (or not selecting any) are reconsidered based on that. Instead of considering A0>2 to identify peaks, we now use a stricter A0>2.8, while for identifying reference stations, we still ask that A0 not exceed 2. The tables are updated accordingly.
- We also added some columns in Table A1 to better analyse the empirical/data-derived results: 1. we add a characterisation of the HVSR shape as 'flat' or 'peaky' in the case where no f0 is chosen. 2. we add a new f0 where it is identified by virtue of the VACF-corrected HVSR (in cases where simple HVSR did not show a clear f0,which is sometimes the case with peaky HVSR). 3. We add a qualitative characterisation of directional sensitivity per frequency range (low, high, very high). 4. We add the number of the clusters as per HVSR and VACF-corrected HVSR. All these better inform our selection of reference site conditions based on seismic data.
- The physical significance of rotational sensitivity is explained better, as well as the metrics introduced to quantify it (SD1-10, SD0.3-30). Indicative values of SD are proposed for characterizing low, high and very high variabilities in HVSR.
- The VACF correction of the HVSR is assigned its own section and explained in much more detail. Figure 9 is added to the main text (it was in the annex before), and it is now updated and improved to include uncertainties. A mosaic plot of all VACF-corrected HVSR across the 60 stations is added to the annex (new fig. A4).
- We stress the limitations of HVSR, that a flat JVSR is not a panacea, and that even the VACF-corrected version of it is not immune to the limitations of the method
- We add a machine-readable excel file as a supplement, containing all tables given in the article to maximize usability of the numerous resulting metadata and parameters.

Moderate/minor changes:

- The abstract was rewritten to better reflect scope and content.
- The initial 60-station rock site selection criteria were better explained. They included many factors used in an inclusive way, and were not based only on belief, so this misunderstanding has now been amended.
- We stress the scientific outcomes and timeliness of the work (for instance, the products of this paper are fully in line with needs identified in the literature, e.g. Zhu et al 2020) and we also plant the work better in context, e.g. in relation to EFEHR, and with Greek and European hazard/risk efforts and models.
- We explain why we do not opt for the data-derived parameters ds2s and k0.
- We explain why we do not present collocated station results for HVSR, and discuss issues of differences at high frequencies based on the suggested literature.
- We explain what we do in cases of double events in the processing.

Editorial changes:

- All individual tables (tables 2,3,4,5,6) are moved to the annex (A1,2,3,4,5) to improve the flow of the article and avoid causing disruption due to their length. We keep 2 tables: one with the basic information of the stations (t1) and another with all criteria are combined to reach a final disposition as to each site's capability as a reference station (old t7, new t2).
- In the final summary table we had omitted the installation column - that is now corrected.
- We now avoid using the term transfer function where it could create misunderstandings.
- All recommended references have been added.

- Added a range of depths in Table 1.
- Improved/clarified figs 1,2 and added letters a and b.
- Improved fig. A1 to allow for colour density to indicate population density.
- Fig 4 is better described in the text.

Not done:

- Although we mentioned we would try, we did not compile DEM data to compute topographic slope for the stations where it was unavailable from external sources. However, as explained in the response and the text too, we feel that the site visits allow for a more accurate assessment of the actual slopes in the vicinity of the station. One of the points we try to make is that we prioritise site-specific operator information to large-scale, poor-granularity data.

We thank you again for all your help and ideas. We hope this new version satisfies your main concerns.

---

## Author Response (AR2)

Dr Olga-Joan Ktenidou
Senior researcher
Institute of Geodynamics
National Observatory of Athens
PO Box 20048, 11810, Athens, Greece
olga.ktenidou@noa.gr

Att.: Dr L. Danciu
Guest Editor, Special issue on European Seismic Hazard and Risk models

Athens,
19/12/2024

Dear Editor,

Please find enclosed the revision of our work entitled 'The quest for reference stations at the National Observatory of Athens, Greece'.

We would like to thank the reviewers as well as yourself for the final minor suggestions and comments. In what follows, we describe the changes made to the manuscript. These are also clearly marked on the annotated documents using 'track changes' and the main changes based on reviewer comments are highlighted in yellow.

Thank you again for your time and consideration of our work. We are looking forward to hearing from you.

Yours sincerely, for the authors,

Olga-Joan Ktenidou

Revisions requested by reviewers (highlighted in yellow in the text):

1. Based on Reviewer #1:
   - "Introduction and dataset" ➔ The $2^{nd}$ paragraph in page 2 is now reworded, as suggested.
   - "Signal processing" ➔ With thanks for the encouragement and appreciation, we have however removed mention to coda windows because they are not analysed in this paper and discussing their potential differences with S-waves would likely sidetrack us.
   - "Data analysis" ➔ The $1^{st}$ paragraph in page 15 is now reworded, as suggested.
   - "Clustering and classification" ➔ Section 3.2 referred to by the reviewer is now called 4.2 because we discovered our section numbering was erroneous. The $1^{st}$ paragraph in page 2 is now reworded, as suggested, and less stress is now placed on defending the lack of ds2s/kappa. Indeed, this long-winded defense was a result of the first review but its effect was distracting.
   - "Discussion" ➔ With thanks for the encouragement and appreciation, we have only made small editorial improvements and clarifications.

2. Based on Reviewer #2:
   - Main remaining comment on VACF: We recognize the point the reviewer is trying to make and the need to stress the tentative nature of the VACF-corrected results. Figure 9 has been moved back to the annex. The $1^{st}$ paragraph in page 17-18 has been reworked in the same spirit, stressing the assumptions and further describing how VACF was initially computed. See also $1^{st}$ paragraph in page 20.
   - All 6 minor comments of editorial nature have been implemented (including amending figures 3 and 9), thank you for catching them.

Additional editorial revisions:

We have carefully re-examined the manuscript and made extensive editorial improvements to the language, our main goal being better clarity, simplicity and succinctness, improved grammar and avoiding some repetitions. We also spotted some issues that needed fixing, namely:

- Our section numbering had errors and repetitions and was fixed.
- Table 1 was lacking its first 2 columns by mistake. This was fixed.
- Figure 4 and 8 were lacking a legend to explain angles of rotation – fixed.
- Figure 9 has more comprehensible data labels.
- A reference was added for the Ward criterion.
- Corrected some affiliations.

---

## Author Response (AR3)

Dr Olga-Joan Ktenidou
Senior researcher
Institute of Geodynamics
National Observatory of Athens
PO Box 20048, 11810, Athens, Greece
olga.ktenidou@noa.gr

Att.: Dr L. Danciu
Guest Editor, Special issue on European Seismic Hazard and Risk models

Athens,
30/1/2025

Dear Editor,

Please find enclosed the final files after editorial corrections of our work entitled 'The quest for reference stations at the National Observatory of Athens, Greece'.

We opted for 2 appendices, A for the tables and B for the figures, so those have been moved from the Supplement. We have also created our Xenodo doi-references repository with an excel file including all tables and downloadable versions of the plots.

We would like to thank you once more for shepherding us through this process.

Yours sincerely, for the authors,

Olga-Joan Ktenidou